# Primary aerosol and secondary inorganic aerosol budget over the Mediterranean basin during 2012 and 2013

Jonathan Guth[1], Virginie Marécal[1], Béatrice Josse[1], Joaquim Arteta[1], and Paul Hamer[2,1]

[1]Centre National de Recherches Météorologiques, CNRS–Météo-France, UMR3589, Toulouse, France
[2]NILU – Norwegian Institute for Air Research, P.O. Box 100 2027, Kjeller, Norway

*Correspondence to:* J. Guth (jonathan.guth@meteo.fr)

**Abstract.** In the frame of the Chemistry-Aerosol Mediterranean Experiment (ChArMEx), we analyse the budget of primary aerosols and secondary inorganic aerosols over the Mediterranean basin during the years 2012 and 2013. To do this, we use a two-year long numerical simulation with the Chemistry-Transport Model MOCAGE validated against satellite and ground based measurements. The budget is presented on an annual and a monthly basis on a domain covering $29°$ North to $47°$ North latitude and $10°$ West to $38°$ East longitude.

The years 2012 and 2013 show similar seasonal variations. The desert dust is the main contributor to the annual aerosol burden in the Mediterranean region with a peak in spring, and sea salt being the second most important contributor. The secondary inorganic aerosols, taken as a whole, contribute to a similar level as sea salt. The results show that all of the considered aerosol types, except for sea salt aerosols, experience net export out of our Mediterranean basin model domain, and thus this area should be considered as a source region for aerosols globally. Our study showed that $11\%$ of the desert dust, $22.8\%$ to $39.5\%$ of the carbonaceous aerosols, $35\%$ of the sulphate and $9\%$ of the ammonium emitted or produced into the study domain are exported. The main sources of variability for aerosols between 2012 and 2013 are weather related variations, acting on emissions processes, and the episodic import of aerosols from North American fires.

In order to assess the importance of the anthropogenic emissions of the marine and the coastal areas which are central for the economy of the Mediterranean basin, we made a sensitivity test simulation. This simulation is similar to the reference simulation but with the removal of the international shipping emissions and the anthropogenic emissions over a $50\mathrm{km}$ wide band inland along the coast. We showed that around $30\%$ of the emissions of carbonaceous aerosols and $35\%$ to $60\%$ of the exported carbonaceous aerosols originates from the marine and coastal areas. The formation of $23\%$, $27\%$ and $27\%$, respectively of, ammonium, nitrate and sulphate aerosols is due to the emissions within the marine and coastal area.

## 1   Introduction

Atmospheric pollution is an environmental problem our modern societies have to face. It has impacts on human health (WHO, 2013), agriculture (Agrawal et al., 2003), ecosystems (Bytnerowicz et al., 2007) and even on buildings (Grossi and Brimble-combe, 2002). It also has an impact on weather and climate (Stocker et al., 2013; Nabat et al., 2015).

The Mediterranean basin region is a region subject to atmospheric pollution, especially for air quality issues (Rodríguez et al., 2006) because of the high population density on the Mediterranean coast. The emission sources are various with most of the anthropogenic and biogenic sources in the northern part of the basin and large mineral dust emissions in the south. The Mediterranean basin also experiences sporadic pollution from forest fires. The accumulation of pollutants is favoured by the synoptic scale flows along with the complex topography of the area. Moreover, the climate simulations tend to show that the climate of the Mediterranean basin will become dryer and warmer, especially during the summer (Giorgi and Lionello, 2008).

In this context, the ChArMEx project aims at acquiring knowledge about the present and the future air chemical composition of the Mediterranean area and aims to understand its various impacts (Dulac, 2014). In the framework of ChArMEx, three intensive observation periods took place in the summer months in 2012 and 2013. In 2012, the TRAQA campaign (transport and air quality above the Mediterranean basin) aimed at characterising the dynamical processes exporting polluted air masses from source regions of the Mediterranean basin. During the TRAQA campaign, 20 June - 13 July 2012, meteorological conditions were mainly favouring continental outflow to the Mediterranean basin from different source regions (Di Biagio et al., 2015). The data collected during this airborne campaign were radiative properties of aerosols (absorption and scattering coefficients), particle number concentration and particle composition. Two intensive campaigns, ADRIMED and SAFMED, were conducted in 2013. The first one, ADRIMED (Aerosol Direct Radiative impact on the regional climate in the MEDiterranean region) took place between the 11 June and the 5 July 2013 (Mallet et al., 2016). The first part of this campaign is characterized by changes in the synoptic flux: easterly (16 June), southerly (19 June), north-westerly (29 June) for example. The ADRIMED measurement strategy was mainly composed of two in situ super-sites (Ersa and Lampedusa) and aircraft and balloon measurements. Super-sites mere measuring aerosol number, cloud condensation nuclei and mass concentrations, aerosol composition and radiation measurements such as absorbing, scattering and extinction coefficients. Aircraft based measurements were composed of aerosol size distribution, number concentration and radiation measurements. The SAFMED campaign (Secondary Aerosol Formation in the MEDiterranean) took place between 24 July and 1 August (Di Biagio et al., 2015). The meteorological conditions during this campaign can be divided into two periods. The first period corresponds to a stable anticyclone located on the western part of the basin until the 26 July possibly causing an accumulation of pollutants in the area. Then, the basin was affected by a cyclonic system on 28-29 July leading to very clean conditions. During the SAFMED campaign, the measured parameters, were similar as those during ADRIMED, being size distribution and particle number concentration and radiative properties.

The aerosol contributions during summer 2012 were analysed by Rea et al. (2015) using the Chemistry-Transport Model (CTM) CHIMERE (Menut et al., 2013). They show that the Euro-Mediterranean region was largely influenced by mineral dust. Indeed, surface $PM_{10}$ were composed of $62\%$ mineral dust while anthropogenic aerosols were the second largest contributor ($19\%$). For $PM_{2.5}$, the anthropogenic emissions were the major part of the surface $PM_{2.5}$ composition ($52\%$). The mineral dust was the second contributor with $17\%$. Biogenic sources also played a significant role in $PM_{2.5}$. This result is consistent with Querol et al. (2009b) that show the importance of desert dust aerosols in the Mediterranean basin. When looking at the Aerosol Optical Depth (AOD), being an indicator of the total column of aerosols, Rea et al. (2015) showed that anthropogenic sources accounted for $34\%$ of the total AOD, while mineral dust for $23\%$ and biogenic sources for $14\%$. Menut et al. (2015) analysed the ozone and aerosol variability between the 1 June and the 15 July 2013. They show this period was not very polluted, mainly

due to several precipitation events. Aerosols in the boundary layer, were dominated by sea salt, sulphate and mineral dust in this case. The column of aerosols was mainly composed of mineral dust.

The past studies focused on the summer season. Here we go a step further by analysing the aerosols over the Mediterranean region based on a two year long simulation that includes the intensive periods (2012 and 2013). Our objective is to establish the budget of the primary aerosols and secondary inorganic aerosols in this region for these two years including an analysis of its seasonal variability. Because particulate pollution is an issue there, we also analyse, using a sensitivity simulation, the contribution of the anthropogenic emissions from in the Mediterranean coast and from international shipping emissions to the aerosol budget. The years 2012 and 2013 having different mean meteorological conditions, this allows us to quantify the impact of this year-to-year meteorological variability on the aerosol budget. With this work being part of the ChArMEx project, the choice of the years 2012 and 2013 for this study was linked to the possible availability of ChArMEx data, such as aerosol composition, on the long term that could be compared to the simulation results. Unfortunately, to day, these data are not available. Nevertheless, these years are of interest because of their different meteorology and also it makes possible to link our results to other ChArMEx studies published in the present issue and to any future studies of this time period. Our study is based on the chemistry-transport model MOCAGE (Josse et al., 2004; Sič et al., 2015; Guth et al., 2016) and the use of a wide range of observations.

The paper is organised as follows. Section 2 presents the MOCAGE model and the simulation set up. The simulation is then evaluated in section 3. In Section 4, we analyse the budget and the variability of the aerosols over the Mediterranean basin. Section 5 presents the results of a sensitivity test aiming at quantifying the impact of the anthropogenic emissions over the Mediterranean sea and its coast. Finally, section 6 concludes this article.

## 2 Configuration of the MOCAGE simulation

This section presents the MOCAGE model used in this study and the set up of the simulation discussed in sections 3 and 4.

### 2.1 The MOCAGE model

MOCAGE (Modele de Chimie Atmospherique à Grande Echelle) is an off-line global chemistry transport model with grid-nesting capability used for research at Météo-France in a wide range of scientific studies on tropospheric and stratospheric chemistry, at various spatial and temporal scales. MOCAGE has been used, for example, for studying the impact of climate on air composition (Teyssèdre et al., 2007; Lacressonnière et al., 2012; Lamarque et al., 2013) or tropospheric-stratospheric exchanges using data assimilation (Barré et al., 2014). MOCAGE is also used for daily operational air quality forecasts in the framework of the French platform Prev'Air (Rouil et al., 2009, http://www2.prevair.org/) and in the European CAMS project (Copernicus Atmospheric Monitoring Service). In CAMS, the MOCAGE model is one of the seven models contributing to the regional ensemble forecasting system over Europe (Marécal et al., 2015, http://macc-raq-op.meteo.fr/index.php).

The version of MOCAGE used in this study is fully detailed in Sič et al. (2015) and Guth et al. (2016). Two chemical schemes are implemented in order to represent both the tropospheric and the stratospheric air composition in MOCAGE. The Regional

Atmospheric Chemistry Mechanism (RACM) (Stockwell et al., 1997) is used in the troposphere. For the stratosphere, it is the REPROBUS scheme (REactive Processes Ruling the Ozone BUdget in the Stratosphere) which is implemented (Lefèvre et al., 1994). Regarding aerosols, the version of the model used in the present study includes desert dust, sea salt, primary organic aerosols, black carbon and secondary inorganic aerosols (SIA) managed through the ISORROPIA module (Fountoukis and Nenes, 2007), including the interactions with sea salt aerosols. The model uses a sectional representation with six size bins for each aerosol type. The sizes are ranging from $2\,\mathrm{nm}$ to $50\,\mathrm{\mu m}$ and there is no effect of ageing on aerosol size in the model concerning aerosol interactions. Yet, aerosol size distribution is affected by deposition and emission or production processes.

The version of MOCAGE model used in this study does not include Secondary Organic Aerosols (SOA). SOA are currently in development in MOCAGE and are not yet validated. In winter, carbonaceous aerosols are mainly composed of primary aerosols from biomass burning and fossil fuel combustion (Gelencsér et al., 2007). Menut et al. (2013) showed by a regional model simulation that in summer, $PM_{10}$ aerosols are dominated by dust and secondary inorganic aerosols over the Mediterranean basin. The fraction of SOA varies between 3 and $16\%$ of the total $PM_{10}$ mass. Yet, SOA can be a significant contributor to aerosols. However, organic aerosol made up about a half of the measured $PM_1$ fine aerosol at Cape Corsica Michoud et al. (2017) during the SOP2 field experiment in summer 2013. Nevertheless, our study analyses the mass budget of aerosol of all sizes in which the fine mode aerosol contribution to total mass is low. From all these studies we can expect that SOA contributions to the mass of total aerosol is small but non negligible and could lead to negative biases compared to observations.

## 2.2 Set up of the simulation

For this study, the model is run using a global domain at $2° \times 2°$ resolution and a nested domain over the Mediterranean basin at $0.2° \times 0.2°$ resolution. This second domain extends from $16°$ North to $52°$ North and from $20°$ West to $40°$ East. The domain simulated is larger than the zone of interest and in order to focus on the basin, we use a sub-domain centred on the Mediterranean basin represented in Fig. 1 by the red square. This domain covers the $29°$ North to $47°$ North latitude and $10°$ West to $38°$ East longitude region and will be called the "budget domain".

The MOCAGE model uses $47$ vertical levels, in $\sigma$-pressure coordinates, from the surface up to $5\,\mathrm{hPa}$. Simulations are run with a spin-up period of 3 months and are driven by the meteorological fields from ARPEGE operational analyses (Courtier et al., 1991).

## 2.3 Emissions

At the global scale, the anthropogenic emissions used are the MACCity emissions representative for 2013 given at a $0.5° \times 0.5°$ resolution (van der Werf et al., 2006; Lamarque et al., 2010; Granier et al., 2011; Diehl et al., 2012). The biogenic emissions are based on Sindelarova et al. (2014) for the volatile organic compounds. They are at a $0.5° \times 0.5°$ resolution, monthly and representative for 2010 . $NO_x$ emissions from the soil come from the GEIA dataset (Yienger and Levy, 1995) while nitrogen oxides from lightning are taken into account following Price et al. (1997). GFAS emissions (Global Fire Assimilation System, Kaiser et al. 2012) giving daily biomass burning emissions based on satellite data are used here. The natural aerosols emissions

are dynamically computed using Marticorena and Bergametti (1995) and Kok (2011) for mineral dust and Gong (2003) for sea salt.

The anthropogenic emissions used at the regional scale are from the MACC-III project emission inventory representative for the year 2011. It corresponds to the latest update of the MACC-II emission inventory (Kuenen et al., 2014). This emission inventory, at a $7 \times 7$km resolution, covers the European continent and the Mediterranean sea. It is completed over the African continent by the MACCity emissions that are also used at the global scale. The other types of emissions are the same as those used at the global scale. In the model, all emissions are distributed on the 5 lowest model layers using an exponential decay with a decay constant of 5. The temporal distribution is based on the monthly variations provided in the inventory chosen on top of which are added the variations linked to the day of the week and the time of the day (following EMEP profiles). The speciation of volatile organic compounds follows Stockwell et al. (1997) based on Middleton et al. (1990).

## 3   Evaluation of the simulation

Before analysing the simulation results, their evaluation has been performed against various observations sources and is presented in this section. Unfortunately, the few ChArMEx measurements over long periods of time were not available at the time of the present study for use of the model evaluation. The statistical indicators used in this section are defined and explained in the Appendix A.

### 3.1   Comparison with MODIS aerosol optical depth

MODIS daily mean AODs were used to evaluate the model simulations. For this purpose, we select both the daily MODIS and Deep Blue data level 3 (L3, collection 6) for the year 2012 and 2013 and perform an additional quality control and screening as presented in Sič et al. (2015) and Guth et al. (2016).

AODs in MOCAGE are calculated at $550\,\mathrm{nm}$ using Mie theory with refractive indices taken from Global Aerosol Data Set (Köpke et al., 1997) and extinction efficiencies derived with Wiscombe's Mie scattering code for homogeneous spherical particles (Wiscombe, 1980).

Figure 1 presents the maps of the annual MNMB (Modified Normalized Mean Bias, dimensionless) of the AOD simulated with the MOCAGE model against the MODIS and the Deep Blue AODs for the years 2012 and 2013. The simulated AOD shows a good agreement with the MODIS AOD with a MNMB close to $0$ in a large area, especially over the Mediterranean sea. The MNMB is slightly negative over the Mediterranean sea for the year 2013, but not for the year 2012. When considering Deep Blue observations, the MNMB is lower, between $-0.5$ and $-1$, over the Red sea, the north of Africa and off the African Atlantic coast, meaning a slight underestimation by the model. The MNMB is higher, between $0.5$ and $1$, north of the Black sea. The negative bias over the North of Africa can be due to an underestimation of desert dust aerosols that may not not transported far enough, and to a lack of secondary organic aerosols, especially for the coastal regions. Indeed, organic aerosols can represent a significant part of the fine mode aerosols (Michoud et al., 2017) and they hence have a noteworthy contribution to AOD in the visible.

Tables 1 and 2 present the statistics for the comparison between the MODIS and Deep Blue AOD data and the MOCAGE simulations for 2012 and 2013 over the whole simulated domain and the budget domain, respectively. They show that the model is able to simulate well the aerosol optical depth over this period and region. The statistical indicators are similar between the two domains considered here. Hence the following discussion will focus on the whole simulated domain comparison (Table 1). There is a slightly different behaviour between the two years. MNMB and FGE (Fractionnal Gross Error, dimensionless) are lower for the 2012 simulation, but the correlation is better for the 2013 simulation.

Table 3 presents the statistics for the comparison between the MODIS AOD data and the MOCAGE simulations for the different complete seasons included in the years 2012 and 2013 over the whole simulated domain. These numbers show a lower correlation in spring that can be related to the desert dust episodes that may be related to a spatial or temporal shift in the model compared to observations that can decrease the correlations. These numbers are consistent with those of Rea et al. (2015) for the summer 2012, despite a lower correlation (0.41 for 2012 in this study versus 0.68). The bias and error metrics do not show a seasonal behaviour. The statistics over the budget domain are very similar to those of the whole simulated domain and are not presented here.

### 3.2 Comparison with AERONET data

AERONET (AErosol RObotics NETwork) measures ground-based AOD from automated stations with an accuracy of $\pm 0.01$ (Holben et al., 1998). The AERONET data are used here for the simulation evaluation as in Sič et al. (2015). In this comparison, we used 33 AERONET stations for 2012 et 40 for 2013.

Figure 2 presents the comparison between the annual aerosol optical depth simulated by MOCAGE and the annual aerosol optical depth measured by the aeronet stations for the years 2012 and 2013. It shows a good agreement. This figure exhibits generally similar patterns on the mean AODs simulated by the model for 2012 and 2013. AODs are highest in 2012 over North Africa and are higher in 2013 than in 2012 over the north-east of the domain, especially over Romania and Ukraine. In the eastern Mediterranean, the AODs are underestimated in 2013. The anthropogenic emissions in this area are not well known and likely underestimated, leading to a systematic negative bias of the model over this region. The difference of behaviour between the years 2012 and 2013 might be due to weather differences. The year 2012 has more rainfall in this area, leading to more wet deposition (Fig. 6). The mean concentrations in 2012, and also the AOD, are lower and closer in the simulation to reality since wet deposition reduces the impact of emission uncertainty. This partly hides the underestimation of the emission inventory compared to 2013. This can also be seen in Fig. 1 when comparing MOCAGE to MODIS and Deep Blue AOD.

Table 4 presents the statistics of the comparison between the MOCAGE simulated AODs and the AERONET observed AODs. The model compares very well to this observation set, with very low MNMB for both years (0.10 and 0.02) and high correlations (0.69 and 0.67). These numbers here are coherent with those of Rea et al. (2015) and Menut et al. (2016).

As for the MODIS AODs, this comparison shows a good agreement between the model simulation and the AERONET AOD measurements. Also, both comparisons reveal coherent patterns such as an underestimation of the modelled AOD over the east coast of the Mediterranean basin.

### 3.3 Comparison with European air quality monitoring stations from the AQeR databases

The description of the data used in this section is available in Appendix B. Table 5 presents the statistics for the comparison between the MOCAGE simulations and AQeR hourly data for the year 2012 and 2013, and for $PM_{10}$ and $PM_{2.5}$. This table shows a similar behaviour for 2012 and 2013. Aerosol concentrations are underestimated, MNMB for $PM_{10}$ are $-0.64$ and $-0.58$, respectively, for 2012 and 2013 with correlations of $0.62$ and $0.59$. $PM_{2.5}$ concentrations are well represented with lower MNMB ($-0.19$ and $-0.27$) and higher correlations ($0.71$ and $0.68$). The aerosol underestimation can be explained, at least partly by the lack of secondary organic aerosols in the model MOCAGE, but also by uncertainties in the anthropogenic emission inventories particularly on the eastern part of the Mediterranean region. The location of the AQeR stations, mostly in the northern part of the simulation domain, makes the influence of natural aerosols very small.

Table 6 presents the same statistics as for Table 5 but over the budget domain. In terms of bias, the results are very similar with a negative bias of about $-10\mu\mathrm{gm}^{-3}$ for $PM_{10}$ and $-4\mu\mathrm{gm}^{-3}$ for $PM_{2.5}$. The errors and the correlation are slightly worse than for the statistics over the whole modelled domain with, for example, a correlation of $0.49$ for $PM_{10}$ in 2012 against a correlation of $0.62$. We should note that for $PM_{2.5}$ there are only 24 and 29 stations in the budget domain for the year 2012 and 2013, respectively. Therefore the statistics are not as solid for the budget domain as for the whole simulation domain.

Table 7 presents the statistics for the comparison between MOCAGE simulations and AQeR hourly data for the different seasons included in the years 2012 and 2013. It shows that the model is in better agreement with the measurements in winter periods compared to summer periods. For example the $PM_{2.5}$ correlation is $0.71$ for the DJF period while it is $0.44$ for the JJA 2013 period, while the MNMB is higher for summer than for winter. This is probably due to the lack of secondary organic aerosols which are mainly produced in summer.

The comparison with AODs, in section 3.1 showed better agreements between the model and measurements with a smaller bias. This can come from a bad representation of the vertical aerosol distribution. For example, if the total amount of aerosols is correct but the aerosols are not concentrated enough in the boundary layer. Also, the size distribution can affect the AOD computation, which is sensitive to the aerosol size, while the $PM_{10}$ indicator is less sensitive to this aspect as long as the aerosols are smaller than 10 microns. Yet, it is difficult to analyse because the AOD and the surface stations do not represent the same quantities. Indeed, the AOD is a measure of the integrated column of aerosol quantities over a large area (for MODIS), while surface stations evaluate the aerosol concentrations at the surface only and on a limited amount of locations. When comparing to AERONET stations, the AOD is also representative of a limited amount of locations, but that are very different from the surface stations ones.

### 3.4 Comparison with the EMEP database

In order to characterise the model behaviour against aerosol composition, we use the EMEP programme measurements (downloaded from the website http://ebas.nilu.no).

From this set of measurements, we use measurements of concentrations of secondary inorganic aerosols, carbonaceous aerosols and sodium and wet deposition of secondary inorganic aerosols and sodium, in order to check the behaviour of the

model regarding these simulated components. We do not present chloride comparisons because the chemistry of HCl with volatile organic compounds is not represented in the model. Gaseous HCl is evaporated when nitrate is condensed on sea salt aerosols.

We only give the results for the year 2013 since there are too few stations available in 2012 to be statistically significant. Nevertheless, results for 2012 on this limited set of data are similar to those for 2013. Figure 3 represents the location of the stations used in this study. This figure highlights the lack of these types of measurements outside Europe. Indeed, the EMEP network only allows us to characterize the north west part of the Mediterranean region. This is a limitation of the comparison.

### 3.4.1 Concentrations measurements

Table 8 presents the statistics for the comparison between the EMEP measurements and the MOCAGE simulation for the year 2013. Secondary inorganic aerosol compounds are slightly underestimated, with MNMBs of $-0.11$ for sulphate, $-0.17$ for nitrate and $-0.19$ for ammonium. Correlations are slightly better for sulphate ($0.58$) than for ammonium ($0.53$) and nitrate ($0.49$). The results presented here are similar to Guth et al. (2016) for the MOCAGE simulations over the whole European continent. This shows the ability of the model to represent the composition of the SIA over the European part of the domain.

The comparison for black and organic carbon aerosols is made over only 7 stations, 6 of which are included in the budget domain. Black carbon aerosol concentrations simulated with MOCAGE are overestimated and exhibit a positive bias of $0.59 \mu g m^{-3}$, with a correlation of $0.66$. Organic carbon aerosols concentrations are underestimated by the model simulations. The bias is of $-2.05 \mu g m^{-3}$, but the correlation is good with $0.66$. The overestimation of black carbon aerosols and the underestimation of organic carbon aerosols can be due to errors on the speciation of the anthropogenic aerosol emissions. Also, part of the underestimation of the organic carbon aerosols can be linked to the lack of secondary organic aerosols in MOCAGE.

Sodium concentrations are compared to 8 measuring station observations, all are located within the study domain. Sodium concentrations are slightly overestimated by the model with a MNMB of $0.31$ and a correlation of $0.47$. Chloride concentrations show a larger overestimation with a MNMB of $0.80$ and a lower correlation of $0.27$.

### 3.4.2 Wet deposition measurements

Table 9 presents the statistics for the comparison between the EMEP wet deposition measurements and MOCAGE simulation for the year 2013. The wet deposition is underestimated for secondary inorganic aerosol compounds, with a MNMB varying from $-0.57$ for sulphate to $-0.28$ for ammonium. This is related to the underestimation of SIA concentrations. The comparison for sodium wet depositions presents an overestimation with a MNMB of $0.36$ for sodium. These results are consistent with the overestimation of sodium concentrations. The high FGE (around $1.40$) and the low correlation for all aerosol compounds show there are large variations between the model and the measurements. Nevertheless, the low MNMBs allow us to be confident in the mean quantity of deposited aerosol.

### 3.5 Conclusion on the evaluation

In this section, we used different sets of observations to evaluate the results of the model. Firstly, we used aerosol optical depth measurements which provide vertically integrated measurements over a large part of the simulated domain. The model shows good results with respect to the MODIS and AERONET observations. When comparing to AERONET data, we show for example very low biases (MNMB of $0.10$ for 2012 and $0.02$ for 2013) and good correlations ($0.69$ for 2012 and $0.67$ for 2013). However, as shown by Michoud et al. (2017) for summer, organic aerosols can represent up to half of the $PM_1$ aerosols. They can then play a significant role for the visible AOD. The fact that the bias is low here can be a sign of compensating errors since the SOA are not taken into account in this study.

Secondly, we compared the simulations to in situ surface observations. The comparison to AQeR database in terms of particulate matter shows a larger bias in summer related to the lack of SOA in the MOCAGE model. This is consistent with the negative bias when comparing the MOCAGE simulation to organic aerosol measurements from the EMEP database. Finally we compared MOCAGE simulation to wet deposition measurements from the EMEP database. We showed a good agreement in terms of bias.

From this evaluation, we show that the MOCAGE simulation give realistic results compared to observations. Nevertheless, there are large regions, especially in North Africa, where we do not have in situ measurements available to evaluate the model.

## 4 Aerosol budget and variability over the Mediterranean basin

The simulation presented and evaluated in Section 3 is now used to characterize the budget of the aerosols over the Mediterranean basin. Note that the information presented concerning the aerosol budget refers to the total aerosol mass.

### 4.1 Methodology

Over the domain considered, using hourly outputs, for a given aerosol species, we define its budget for a chosen time period by the equation:

$$\Delta_{burden} = Em + Pr - Loss - Dep + Tran, \tag{1}$$

with $\Delta_{burden}$ the difference of the atmospheric burden between the end and the beginning of the time period. $Em$ is the emission, $Pr$ the chemical production, $Loss$ the chemical loss, $Dep$ the deposition terms (dry and wet deposition, and sedimentation) and $Tran$ the import/export of the aerosol in the budget domain. This last term is positive when aerosols are imported into the domain and negative when exported. All terms are estimated or directly computable from the simulation outputs. For the advection, the MOCAGE model uses a semi-Lagrangian transport scheme. It means that for each model grid point, the transport over a time-step is done by determining the location from which the air mass originated at the beginning of the time-step and the associated concentration of aerosol species at this location. This approach is used in order to be able to use long time-steps for the transport. For the MOCAGE model, the transport time-step is set to one hour. Because of the use of a semi-Lagrangian approach in MOCAGE, the $Tran$ term cannot be directly estimated since there is no Eulerian flux

computed in the transport scheme. We therefore use an indirect estimation of the $Tran$ term by calculating the difference of the burden before and after the transport into the budget domain at each time-step. Note that the separation between the inward flux and the outward flux of transported particulate matter cannot be done in the $Trans$ term.

From this definition, the $Trans$ term implicitly includes the transport but also the model errors due in particular to possible mass imbalance. Since mass conservation is insured at the global domain that serves to force the boundaries of the regional domain, the model error due to mass imbalance is expected to be small compared to transport.

Using all these terms we calculate the residual mass, corresponding to the model error that is computed using:

$$Resid = Em + Pr - Loss - Dep + Tran - \Delta_{burden}. \tag{2}$$

We have calculated this residual model error term. It is about $1\%$ of $Em$ or $Pr$ for black carbon and primary organic carbon and sulphate, about $4\%$ of $Em$ for desert dust and sea salt, and about $0.1\%$ of $Pr$ for ammonium and nitrate. Therefore it is small and does not affect our budget analysis. When summing up the budget terms, the reader will then find that the budget is not fully closed. This is due the error made in the semi-lagrangian transport scheme that has a smoothing effect on the strong peak in aerosols concentrations usually leading to a gain of mass, but this errors remains small.

## 4.2 Results of the aerosol budget over the Mediterranean basin

In this subsection, we present the aerosol budget over the two year period 2012-2013, on an annually and monthly basis in order to discuss the seasonal variability. All the following results are presented on the budget domain covering 29° North to 47° North latitude and 10° West to 38° East longitude over the entirety of the vertical extent of the model. All the source terms are positive (emission, production), while the sinks are negative (deposition, chemical destruction). The horizontal transport term is positive for import in the domain and negative for export.

### 4.2.1 Annual budget

Tables 10 and 11 present the annual budget, on the budget domain, of the aerosols for the year 2012 and 2013, respectively. Note that the unit of the burden term is Tg while the other term's unit is Gg. For ammonium, nitrate and sulphate there are no emissions in the model. The first column corresponds to the quantity of secondary aerosol condensed. In the same way, the column "chemical loss" shows the evaporation of the secondary aerosols. One can see the similar behaviour for both years, and especially for black carbon. Desert dusts and sea salt are the most abundant aerosols in the region with a burden of about $900$Gg for desert dust and $150$Gg for sea salt. Other aerosols have a mean burden between $13$ and $75$Gg. But altogether the different SIA components (Secondary Inorganic Aerosols, being sulphate, nitrate and ammonium) add up to a burden similar to sea salt. The year 2012 shows higher concentrations of primary carbonaceous aerosols and secondary inorganic aerosols, while the year 2013 is characterized more by natural aerosols (desert dust and sea salt). For both years, one can see that all types of aerosols experience net export out of the budget domain, except for sea salt. One can also note that the chemical destruction term for the sulphate is equal to zero. This is because ISORROPIA assumes all of the sulfuric acid is condensed into the aerosol phase whatever the thermodynamic conditions are because of the very low vapour pressure of sulfuric acid.

Tables 12 and 13 present the annual budget of the aerosols for the year 2012 and 2013, respectively, as a percentage of the emission or the production. This allows us to easily identify which proportion of the aerosol goes preferentially to each term of the budget. Aan de Brugh et al. (2011) found that wet deposition is the major sink for carbonaceous aerosols, ammonium, nitrate and sulphate aerosols, while desert dust and sea salt experiences mainly dry deposition. Consistently, here wet deposition

is the major sink for primary organic carbon while desert dust and sea salt are mainly deposited by dry processes. This is due to the size of the emitted aerosol which is larger than for the other types of aerosols. The sedimentation is thus more effective. For ammonium and nitrate, the main sink is evaporation which is not described as a separate term of the budget in Aan de Brugh et al. (2011). Wet deposition is the main sink in Aan de Brugh et al. (2011) for sulphate, but sedimentation and dry deposition are the main sinks in this study. This can be explained by the difference of the simulated domains in both studies. The domain

in our study is more southern and has high sulphate aerosol concentrations in the eastern part of the basin associated with less precipitation. For black carbon aerosols, the main sink in our study is export. The difference between the two studies can be explained, similarly to sulphate aerosols, to the difference of domain location and weather conditions, especially in the eastern Mediterranean.

Concerning the export, we can note that $11\%$ of the desert dust is exported, while this percentage raises between $22.8\%$

to $39.5\%$ for the carbonaceous aerosols and the sulphate, $9\%$ for the ammonium and $2\%$ for the nitrate. For the sea salt it is $0.7\%$ for 2012 and $2\%$ for 2013, but the residual mass in the budget calculation is of the same order as the $Tran$ term. We can then consider the global behaviour of sea salt as if there is almost no flux. Nitrate aerosols export is low compared to both ammonium and sulphate. It can be explained by the fact that nitrate and sea salt are linked by the ISORROPIA module. Indeed, with our domain being largely over a maritime surface, there are a lot of sea salt aerosols on which nitrate condenses rapidly

due to the thermodynamic equilibrium assumption. Im et al. (2014) conducted an analysis of the budget of pollutants, including aerosols, over Europe for the year 2008. They found that horizontal advection is a sink for nitrate, ammonium, sulphate and carbonaceous aerosols. Hence, our study is consistent with their results. These results are also consistent with Aan de Brugh et al. (2011).

Guo et al. (2016) showed that the nitrate aerosol partitioning is sensitive to the pH. The pH is taken into account in the

25 ISORROPIA calculation using ions concentrations and the water content as input. Although the pH has not been validated extensively in our model, Guo et al. (2016) showed the pH is well predicted by ISORROPIA. The water content input is provided by the numerical weather prediction simulation, in which data assimilation is made. Therefore, we assume the input information is as precise as possible. Moreover, the use of alkaline components in the SIA computation might change the results. A work is in progress to include it into the ISORROPIA implementation in MOCAGE. The inclusion of these compounds would

increase pH and shift the partitioning of HNO3 to the aerosol phase. Hodzic et al. (2006) showed that the inclusion of dust related alkaline components improves the behaviour of the model, over Europe, by reducing the $PM_{10}$ bias of 6 to $10\%$ when comparing results to selected EMEP stations. By working around the Mediterranean Basin, we expect a greater impact due to the close desert dust sources inducing more frequent and more important dust outbreaks over the domain. Moreover, Ansari and Pandis (1999) showed the bias of the total nitrate aerosol is reduced by about $20\%$ when including crustal species into the

computation. Nevertheless, Fairlie et al. (2010) pointed out that the uptake coefficient usually used are too important. Hence the previous numbers might overestimated.

To further facilitate the analysis of the import/export of the aerosols, Figs. 4 and 5 present the yearly mean of the total column of the different aerosols for the year 2012 and 2013, respectively. These panels also present the red square representing the budget domain. Figure 6 depicts the precipitation rate and the wind fields at 200m above the surface for the years 2012 and 2013. One can see the mark of the desert dust emissions along the southern boundary of the domain (top left panel). These emissions are transported with the dominant Easterly and North-Easterly winds, thus explaining the general export behaviour of the desert dusts. We observe the same phenomenon for carbonaceous aerosols and sulphate whose concentrations are at a maximum in the eastern part of the basin and associated with Westerly winds, which export the aerosols across the eastern boarder. The high concentrations of sulphate in the Eastern Mediterranean are consistent with Dayan et al. (2017) who conducted a study about the aerosol budget over Europe for the year 2006. Dayan et al. (2017) explain this high particulate sulfate abundance on the eastern Mediterranean basin by the westerly winds transporting sulphur rich air mass from central Europe and the intense solar radiation enhancing the $SO_2$ conversion.

By comparing the precipitation rates and the wind fields for 2012 and 2013, we can note differences in the meteorology between the two years. In the Western part of the basin, the year 2013 presents higher wind speed values, especially over the gulf of Lion and North Africa. This explains the higher values of desert dust and sea salt aerosols in this region in 2013 compared to 2012. On the Eastern part of the basin, the year 2012 presents higher wind speed values over the sea, explaining here higher sea salt aerosol levels. Desert dust presents higher total column concentrations in 2013 and a bigger extent towards the north-east of the domain. Associated with lower precipitation in the area between Tunisia and Turkey in 2013, it explains the lower wet deposition of desert dust aerosols in 2013 compared to 2012 despite larger emissions. In this region, we can also see higher concentrations of carbonaceous aerosols in 2012, which can be explained by higher speeds of the wind advecting the pollution from the coast of Aegan sea over the basin.

### 4.2.2 Monthly budget

In this subsection, we examine the aerosol budget at the monthly temporal scale. Figure 7 presents the monthly variations of budgets for the primary aerosols while Fig. 8 shows the monthly variations of budgets for the secondary inorganic aerosols. In Fig. 7, sea salt aerosols present very similar monthly variations between the two years. Sea salt aerosols show a slight annual cycle with more emissions in the winter months, which are related to higher wind speeds. Desert dusts have a similar behaviour between both years, with high levels of dust between January and June and lower dust levels during the second half of the year. Nevertheless, we can note the large differences of desert dust emissions between the year 2012 and 2013, also seen on other budget terms. This illustrates the important inter-annual variability that can be seen over the Mediterranean basin (Querol et al., 2009b). The 2012 dust period starts and finishes earlier, while the dust period in 2013 involved higher emissions, and thus more deposition, export and burden. For 2013, we can explain this phenomenon by looking at the winds during the more active dust season (not shown). In 2013, the average low level winds were stronger over North Africa, leading to higher desert dust emissions, and thus to higher values for all the terms of the budget.

Concerning anthropogenic aerosols, black carbon presents a very similar behaviour between the two years, with a slight annual cycle having higher emissions in autumn and winter than in summer. This is consistent with the monthly variations of the emission inventory used. Organic carbon presents also a similar behaviour for both years, except in summer when there is a higher or similar burden in 2013 despite lower emissions. This comes from the import of aerosols from outside the budget domain. Figure 9 presents the total column and the biomass burning emissions for July 2012 and 2013. This panel illustrates that there were more fires during summer 2013 in North America compared to summer 2012. These fires exported a large amount of aerosols from the North American continent into the budget domain, explaining the difference of behaviour for organic carbon aerosols in summer between 2012 and 2013.

The budget for secondary inorganic aerosols is presented in Fig. 8. There is a similar behaviour for all secondary inorganic aerosols for both years. Nitrate and ammonium show small seasonal variations. The burden of sulphate aerosols has a strong annual cycle with maximum burden in summer despite a lower production during this season. The summer maximum for sulphate aerosols is consistent with Querol et al. (2009a), saying that the Eastern Mediterranean basin is influenced by Eastern Europe countries favouring sulfuric acid formation. The reason for the increase of the burden is the lower deposition in summer than in winter.

### 4.2.3   Conclusions on the aerosol budget

To conclude this section on the aerosol budget over the Mediterranean basin, we highlight several points. Firstly, there are very large differences in the atmospheric loading of the different aerosol types. The burden of desert dust and sea salt is much higher than that of other aerosols, but not only in summer, as already shown by Rea et al. (2015) and Menut et al. (2015), but also throughout the whole year, leading to their predominance in the annual budget. The use of the results of the two simulated years allowed us to observe inter annual differences, with the year 2012 having higher anthropogenic aerosol concentrations, while the year 2013 presents higher natural aerosol concentrations. Secondly, we saw that while dry deposition processes are the main sink for natural aerosols and sulphate, wet deposition is the main sink for primary organic carbon, transport (export) for black carbon and aerosol evaporation for ammonium and nitrate.

We find that all aerosols are exported on average from the domain of study, except for sea salt. Aan de Brugh et al. (2011) and Im et al. (2014) studies about the European aerosol budget in 2006 and 2008, respectively, showed that Europe is a net exporter of anthropogenic aerosols. These results are not directly comparable to our study because of the difference of simulated domain, but it allows some confidence in our results. Based on concentration and wind maps, we showed that desert dust aerosols are exported out of the study domain through the southern boundary, while carbonaceous aerosols and sulphate are mainly exported out of the study domain through the eastern boundary.

The monthly budget shows an annual cycle that is more (desert dust) or less pronounced (sea salt, carbonaceous aerosols) depending on the type of aerosols. This annual cycle is mainly due to the annual cycle of the emissions. It can be related to the weather conditions, influencing directly the amount of emitted particles (desert dust and sea salt). Forest fire emissions from other continents, inducing aerosol import in the Mediterranean basin, are another source of variability for the aerosol budget. It is especially the case for the primary organic carbon in summer 2013.

## 5 Sensitivity study: impact of international shipping and coastal anthropogenic emissions

The Mediterranean basin has a large population density in the coastal areas and high maritime traffic associated with high harbour economic activity linked to the shipping business. In this section, we assess the impact of the anthropogenic emissions in the coastal area and the international shipping emissions in the Mediterranean basin. To address this, we made a second simulation where we removed the anthropogenic emissions over the sea and over a 50km wide band along the coast. All the other parameters of the simulation remain the same. Figure 10 presents the mask used to remove the anthropogenic emissions for this sensitivity test along with the budget domain. in red that is the same as in Section 4. Since the natural aerosols are not impacted by the changes made, we will not include them in this analysis.

Tables 14 and 15 present the annual budget for the sensitivity simulation for 2012 and 2013, respectively. Concerning the black carbon aerosol, we can note a similar behaviour between the two simulated years, which is consistent with what we found in Section 4. Concerning primary organic carbon, we can still see the impact of the biomass burning from North America in summer 2013 on this aerosol. Secondary inorganic aerosols present a similar behaviour between the two years.

In order to compare the results between the two simulations, Table 16 and 17 present the relative differences, between the reference simulation and the test simulation, respectively for the year 2012 and 2013. These relative differences, for the parameter A, are computed as follows:

$$A_{diff} = \frac{A_{sen} - A_{ref}}{A_{ref}}, \qquad (3)$$

with $A_{ref}$ the value of the parameter A in the reference simulation and $A_{sen}$ the value of A in the sensibility test simulation. A negative value means the parameter is smaller, in absolute value, in the test simulation than in the reference simulation. The reader may find some small differences while computing the terms of these tables because they are computed with the real values while the numbers in the tables are rounded. As for the import/export terms, the values are always negative or null, a negative value of the difference means the export is less pronounced in the test than in the reference.

For the black carbon aerosols, we see a similar behaviour between 2012 and 2013. The mean burden is reduced by about 17% while the emissions are reduced by 30% and the export by 35% (for 2013) to 40% (for 2012). This is due to the high black carbon emissions in the eastern part of the domain in the highly populated areas near the coast that are largely exported. Primary organic carbon aerosols have a mean burden reduced by 7.5% while the emissions are reduced by 27% and 29%, respectively, for 2012 and 2013. Here we can see the impact of the high aerosol concentrations coming from the biomass burning in North America. The differences in the imported term for each year, between the reference simulation and the test simulation, are very similar with 0.15Tg for 2012 and 0.14Tg 2013. This represents the reduction of the export of aerosols from local sources. The reference simulation presents an export of 0.32Tg for 2012 and 0.24Tg for 2013. Then, when calculating the relative difference, the year 2013 gives a higher number.

Concerning SIA, both years are similar with a decrease in the mean burden of about 16% for ammonium, 12% for nitrate and 17% for sulphate. The decrease in SIA formation is between 23.2% for ammonium and 36.4% for sulphate and the export decreases of 55% for sulphates and 54% for ammonium. Figure 11 presents the total annual emission for the SIA precursors, computed over the budget domain, in the reference simulation and the sensitivity test simulation. This figure presents the

numbers for 2013, but they are very similar for the year 2012. We can see that the $SO_2$ emissions are reduced by approximately 40%, which is coherent with the 36.4% sulphate formation decrease. Marmer and Langmann (2005) shows that 54% of the summertime sulphate aerosol burden over the Mediterranean originates from ship emissions. In our study, the estimation of the proportion of sulphate aerosols originating from maritime and coastal emission is lower because our proportion is calculated

for the annual quantity. The decrease of the $NO_x$ emissions is about 50%, while the formation of nitrate aerosol is lowered by only 26.7%. The precursor of nitrate aerosols is nitric acid but there are different chemical pathways $NO_x$ can take, explaining the difference between the $NO_x$ emission reduction and the nitrate formation decrease. Moreover, the decrease of sulfuric and nitric acid can also lead to a change in the aerosol pH and then for the aerosol partitioning. Ammonium formation is lowered by 23.2% while ammonia emissions are only lowered by about 20%. This is explained by the fact that ammonium is condensed

onto sulphate and nitrate particles to neutralize the solution. The decrease of sulphate and nitrate becomes then a limiting factor for the formation of ammonium aerosols.

    As a conclusion we can note the high importance of the coastal area, which includes many major cities, in this region. Indeed, our sensitivity test shows that 23% (ammonium) to 36% (sulphate) of the emission or production of anthropogenic primary aerosols and secondary inorganic aerosols in the Mediterranean are originated from the marine or coastal area. Also,

they account for 35% (black carbon) to 90% (nitrate) of the exported aerosols outside the budget domain. We do not show the monthly budgets here since they do not give additional information.

## 6   Conclusions

This study aimed at establishing the budget of the primary aerosols and secondary inorganic aerosols on the Mediterranean basin based on numerical simulations of the years 2012 and 2013 using the MOCAGE model. We also studied its seasonal vari-

ability, its year-to-year variability with meteorological conditions, and the contribution of local anthropogenic emissions from the populated coastal area and from international shipping in the Mediterranean basin. Firstly, we compared the simulation to observations in order to do an evaluation of the simulation. We showed the model was able to well represent the aerosol optical depth on the Mediterranean basin using MODIS, Deep Blue and AERONET data. Secondly, we compared the model simulations to in situ concentration data from the AQeR and EMEP database. This comparison shows that the model represents well

the secondary inorganic aerosols while the lack of secondary organic aerosols is clearly apparent both on seasonal particulate matter observations and on aerosol composition observations. Also, this comparison highlights a lack of observations in the southern part of the domain, and especially in North Africa to fully evaluate the model using in situ surface measurements.

    Secondly we use the two year-long simulation to compute the aerosol budget over the Mediterranean basin on a annual and monthly basis. The budget domain we chose aimed at capturing the economic activities over the Mediterranean sea and the

associated harbour activities while being rectangular for simplicity of treatment and analysis. The two years simulation allowed us to illustrate the inter annual variability of the aerosol budget. While the year 2012 presents more anthropogenic aerosols, the year 2013 has more natural aerosols. We showed that all aerosols considered in this study, except for sea salt, experience net export of our domain of study. Hence this area should be considered as a source region for these aerosols. These results

are consistent with Im et al. (2014) and Aan de Brugh et al. (2011). For desert dust; this result strongly depends on the domain used to do the calculations. We showed that the export of desert dust out of the study domain is due to the position of the southern border including desert dust emissions associated with north-east winds exporting this emissions out of the domain. For the other aerosols, the results are more robust to the location of the limit of the domain used. Our study showed that $11\%$ of the desert dust, $22.8\%$ to $39.5\%$ of the carbonaceous aerosols, $35\%$ of the sulphate and $9\%$ of the ammonium emitted or chemically produced into the study domain are exported.

We observed an annual cycle for the natural aerosols budget that is due to the influence of meteorological conditions, modulating the emissions of desert dust and sea salt. The annual cycle can also be affected by the differences in primary anthropogenic aerosol emissions or variations on the import of aerosols from outside (especially biomass burning events). We also show that natural aerosols (desert dust and sea salt) are predominant over this region all over the year, such as found in Rea et al. (2015) and Menut et al. (2015) for the summer of 2012 and 2013.

In this study we did not include crustal species interactions in the computation of the SIA which can have an important impact (Hodzic et al., 2006; Ansari and Pandis, 1999). It would be interesting to add these interactions because they change the size distribution of the SIA, hence the budget will be change through the different physical processes, such as sedimentation which is sensitive to the aerosol size.

Then, we made a sensitivity test to assess the importance of the marine and coastal regions of the Mediterranean basin. To do this, we removed the international shipping emissions and the anthropogenic emissions over the sea and over a $50\mathrm{km}$ wide band along the coast. We showed that around $30\%$ of the emissions of carbonaceous aerosols and $35\%$ to $60\%$ of the exported carbonaceous aerosols originates from this region. The formation of $23\%$, $27\%$ and $27\%$, respectively, of ammonium, nitrate and sulphate aerosols is due to the emissions within the marine and coastal area. We showed non linear interaction between ammonium and nitrate aerosols and their precursors, as the decrease of their formation does not follow the precursor emissions decrease as it is the case for sulphate aerosols.

The focus of this study is on primary aerosols and secondary inorganic aerosols. Once the SOA development is validated in MOCAGE, it would be interesting to do the budget for this type of aerosol too. Also, this would give an opportunity to analyse the gaseous phase compounds and their budget over the Mediterranean basin. This study will also use the sensitivity test simulation to compare the differences in behaviour between aerosols and gaseous compounds.

We showed in section 4 differences in the average meteorology between 2012 and 2013, and a direct link between the weather conditions and the aerosol concentrations, such as the effect of the wind speed. To go a step further, we propose in a future paper to analyse the aerosol distribution using a more detailed meteorological analysis, based on the concept of weather regimes. Weather conditions can be classified into weather regimes that correspond to idealized meteorological situations. These weather regimes can be used to gather similar meteorological conditions and to analyse the "aerosol regime" associated to each weather regime. This kind of methodology could also be used in climate simulations to assess the expected behaviour of aerosols in the future.

## Appendix A:  Metrics used for evaluation

Several statistical indicators can be used for model evaluation against in situ data. Seigneur et al. (2000) state that past model performance evaluations have generally used observations to normalize the error and the bias. This approach can be misleading when the denominator is small compared to the numerator. Following Seigneur et al. (2000), we chose to use the fractional

bias and the fractional gross error instead of the bias and the root-mean-square error (RMSE).

The fractional bias, also called Modified Normalized Mean Bias (MNMB) or mean fractional bias (MFB), used to quantify, for $N$ observations, the mean between modelled ($f$) and observed ($o$) quantities is a dimensionless quantity, defined as follow:

$$\text{MNMB} = \frac{2}{N} \sum_{i=1}^{N} \frac{f_i - o_i}{f_i + o_i} \tag{A1}$$

The fractional bias ranges between $-2$ and $2$ varying symmetrically with respect to under and overestimation.

The Fractional Gross Error (FGE), also called mean fractional error (MFE) aims at quantifying the model error. It varies between $0$ and $2$ and is a dimensionless quantity, defined by:

$$\text{FGE} = \frac{2}{N} \sum_{i=1}^{N} \left| \frac{f_i - o_i}{f_i + o_i} \right| \tag{A2}$$

The correlation coefficient $r$ indicates the extent to which patterns in the model match those in the observations and is defined by:

$$r = \frac{\frac{1}{N} \sum_{i=1}^{N} \left( f_i - \overline{f} \right) \left( o_i - \overline{o} \right)}{\sigma_f \sigma_o} \tag{A3}$$

Where $\sigma_f$ and $\sigma_o$ are standard deviations, respectively, from the modelled and the observed time series and $\overline{f}$ and $\overline{o}$ their mean values.

## Appendix B:  Description of the AQeR database

A dense measurement network is used for air quality monitoring in Europe. Data are gathered into a database named AIRBASE.

It is managed by the European Topic Centre on Air Pollution and Climate Change Mitigation on behalf of the European Environment Agency (EEA). AIRBASE data are used in this study to evaluate the performance of the model for $PM_{10}$ and $PM_{2.5}$. From 2013, EEA changed their observation database which is now called Air Quality e-Reporting (AQeR).

For this study, we use the latest version (version 8) of the AIRBASE database for the year 2012, and the AQeR database for the year 2013. For simplicity, we will use AQeR to designate both databases.

Monitoring stations from the AQeR database are located on various sites being representative of rural, peri-urban or urban conditions. For model evaluation, we select the stations which are representative of the model resolution. Following Joly and

Peuch (2012), each station is characterized by a class between 1 and 10 according to its statistical characteristics. Classes 1 and 2 corresponds to a fully rural behaviour while 9 and 10 to a traffic behaviour. Then, as in Guth et al. (2016), only the stations corresponding to classes 1 to 5 are kept in order to assure a set of representative sites.

*Acknowledgements.* We would like to thank the Chemistry-Aerosol Mediterranean Experiment project(ChArMEx , http://charmex.lsce.ipsl.fr),
5 which is the atmospheric component of the French multidisciplinary program MISTRALS (Mediterranean Integrated Studies aT Regional And Local Scales). ChArMEx-France was principally funded by INSU, ADEME, ANR, CNES, CTC (Corsica region), EU/FEDER, Météo-France, and CEA. This work has been possible thanks to the AIRBASE, EMEP database and the EBAS database infrastructure. We also acknowledge the MODIS mission team and scientists for the production of the data used in this study. The authors would also like to thank the AERONET PIs and their staff for establishing and maintaining the sites used in this investigation.

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

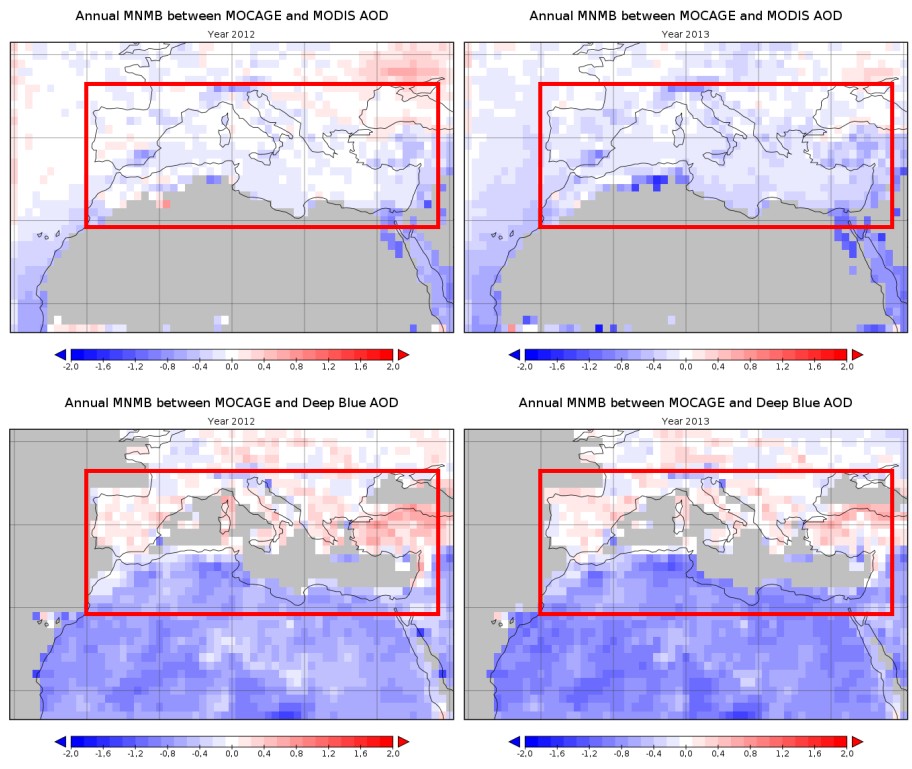

**Figure 1.** Map of the annual modified normalized mean bias (MNMB) of the aerosol optical depth against MODIS observations for the year 2012 (left) and 2013 (right) for MODIS (top) and Deep Blue (bottom).

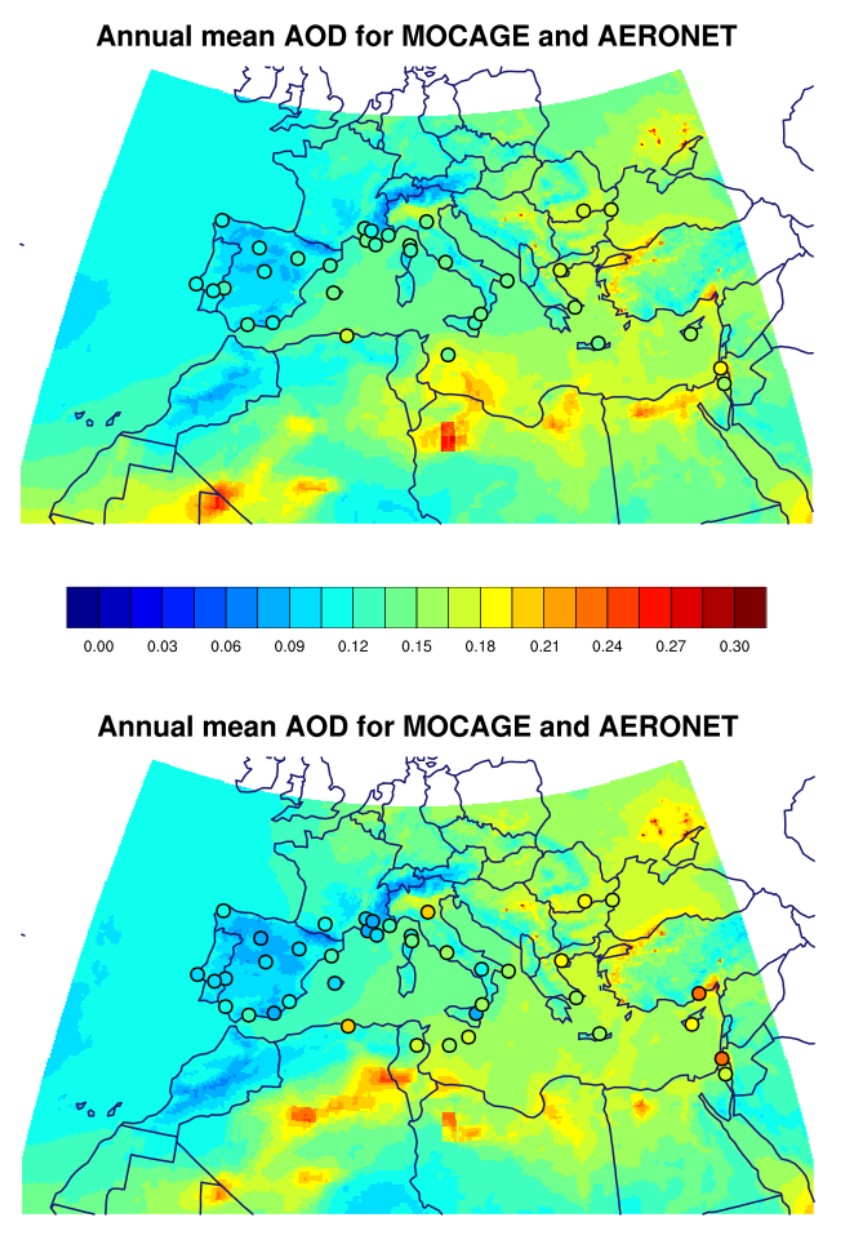

**Figure 2.** Map of the annual mean aerosol optical depth simulated with the model MOCAGE with superimposed AERONET observations (circles) for the years 2012 (top) and 2013 (bottom).

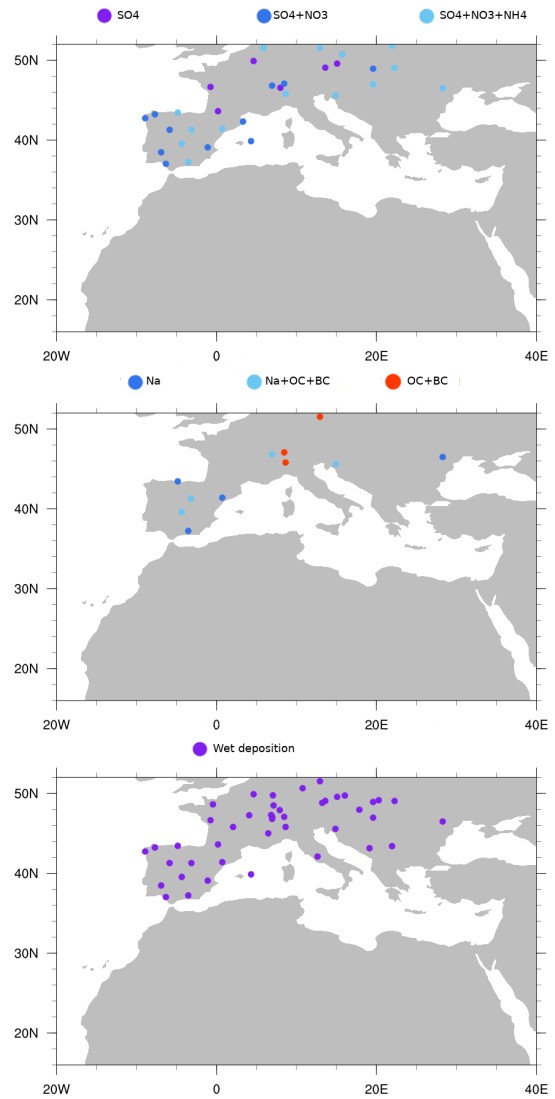

**Figure 3.** Location of the EMEP stations used in this study for the year 2013.

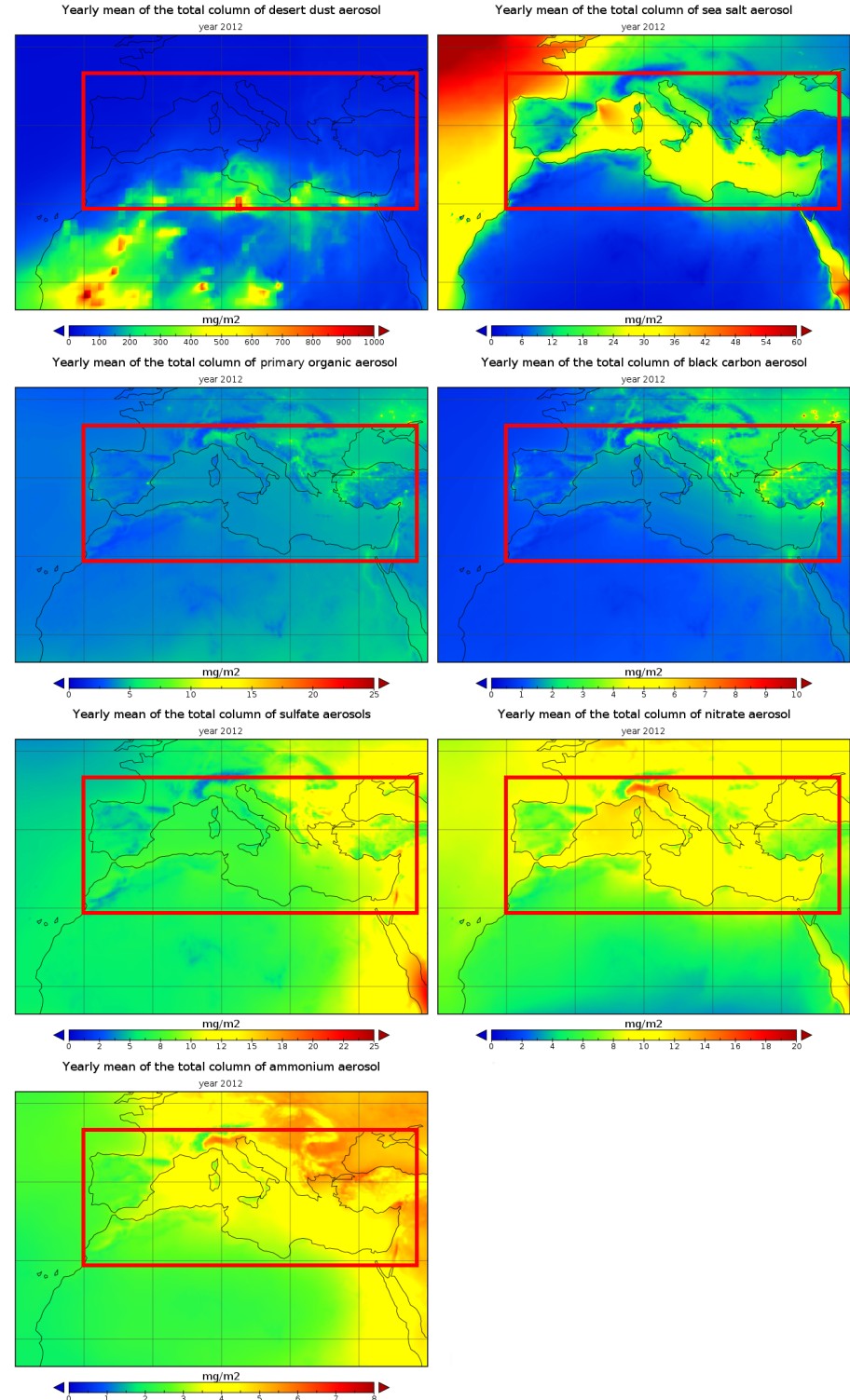

**Figure 4.** Yearly mean of the total column of aerosols for the year 2012. The red square on the figures represents the budget domain.

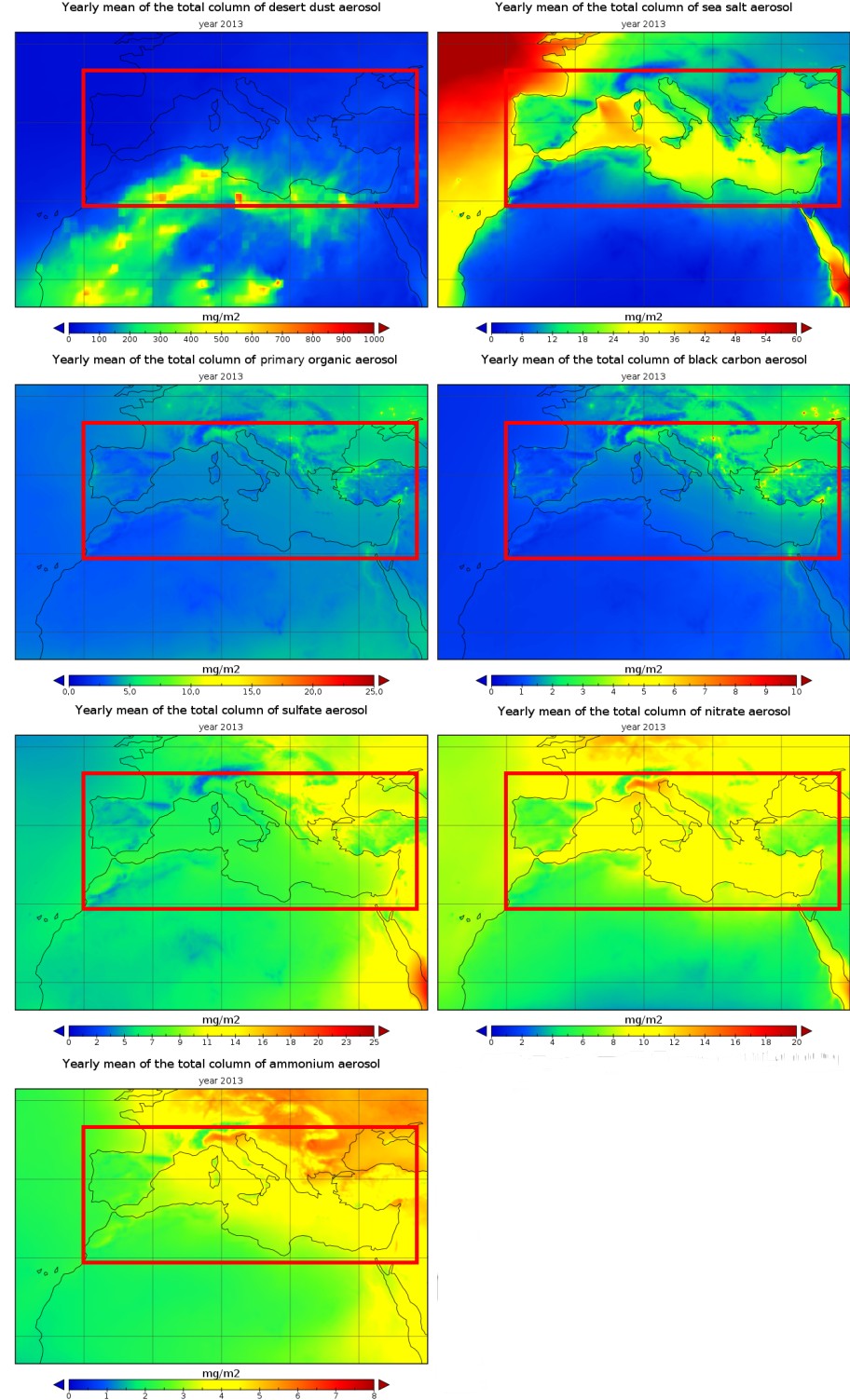

**Figure 5.** Yearly mean of the total column of aerosols for the year 2013. The red square on the figures represents the budget domain.

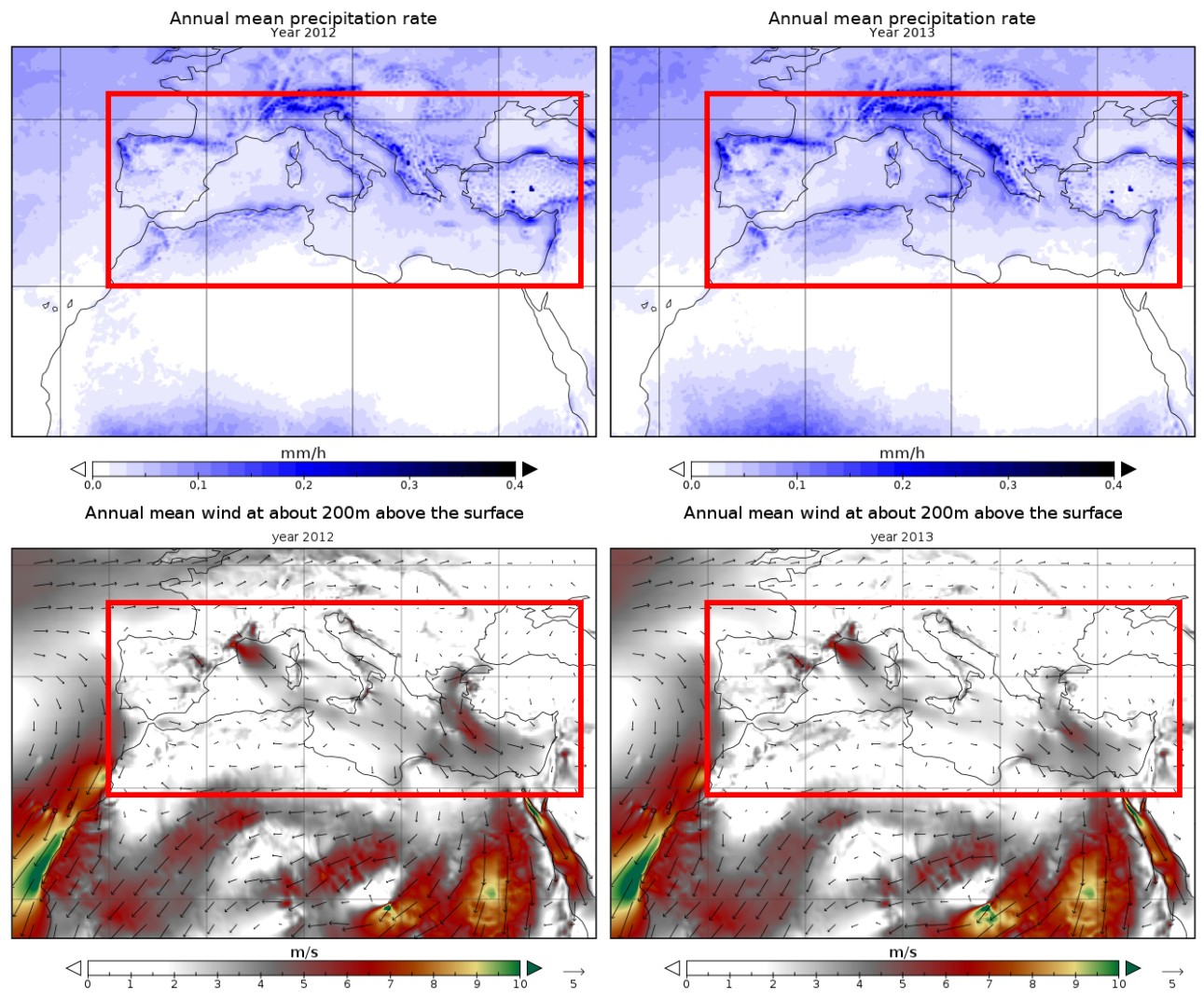

**Figure 6.** Yearly mean of precipitation rate (top panels) and wind vectors at 200m above the surface (bottom panels) for the year 2012 (left) and 2013 (right). The red square on the figures represents the budget domain.

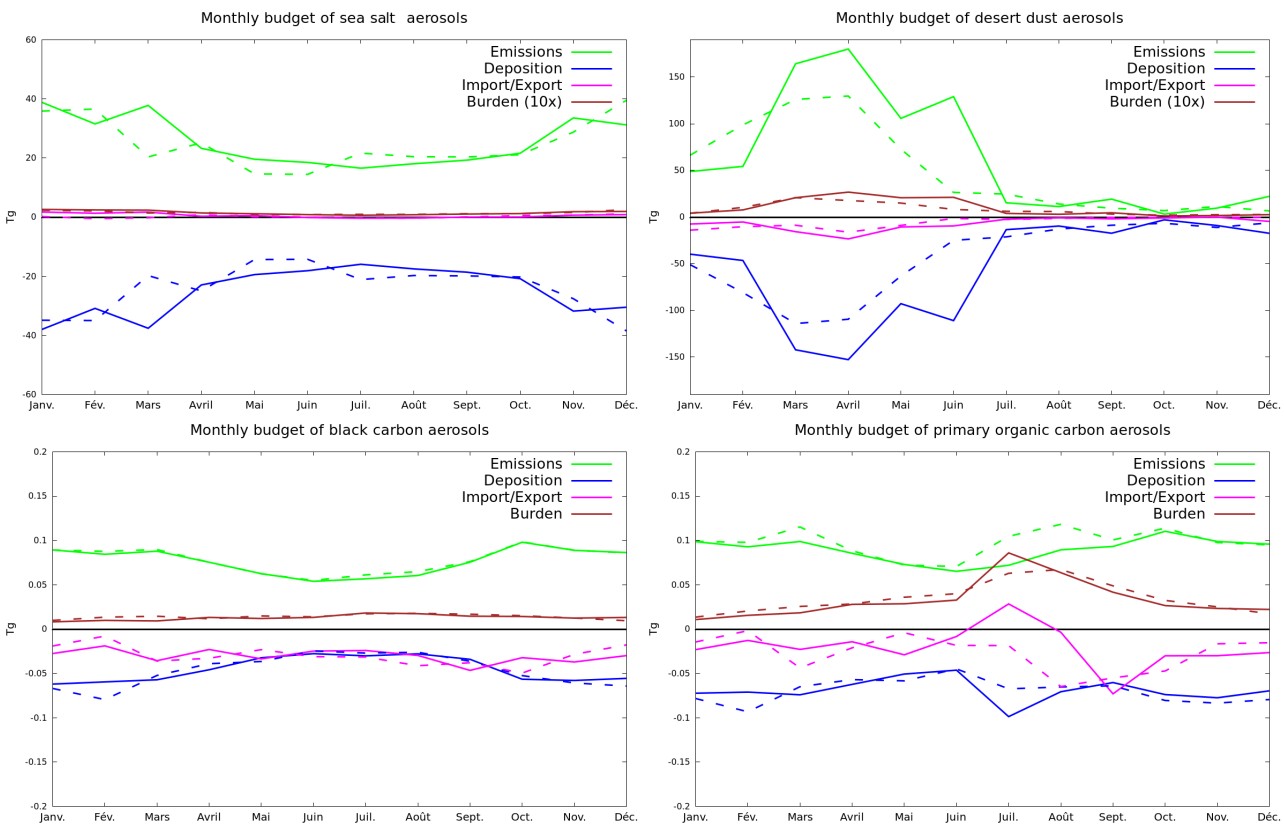

**Figure 7.** Monthly budget for the total mass of primary aerosols for the year 2012 (dashed lines) and 2013 (solid line). The green lines correspond to the emissions, the blue ones to deposition, the pink ones to the import/export part and the brown ones the burden.

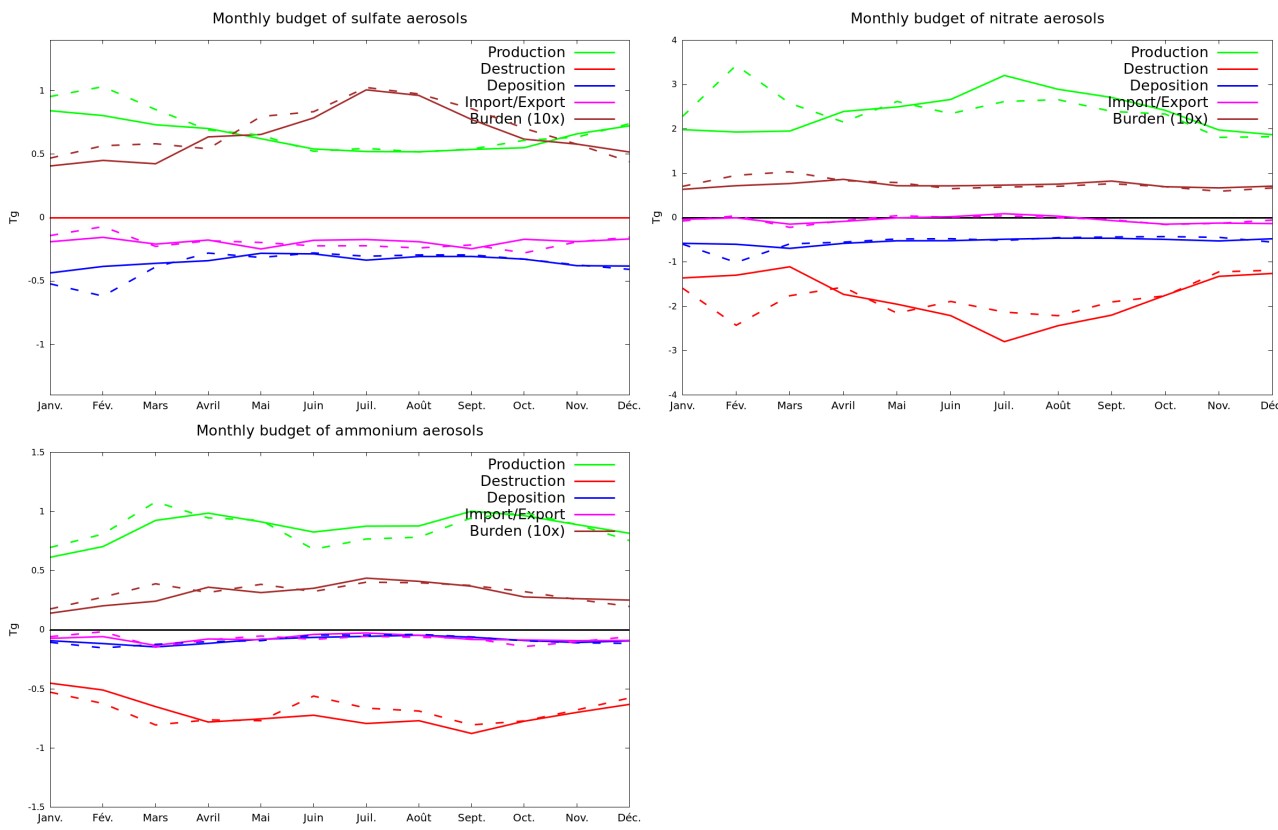

**Figure 8.** Monthly budget for the total mass of secondary aerosols for the year 2012 (dashed lines) and 2013 (solid line). The green lines correspond to the emissions, the red ones to the chemical loss, the blue ones to deposition, the pink ones to the import/export part and the brown ones the burden.

Daily mean organic carbon emissions from biomass burning
July 2012

Daily mean organic carbon emissions from biomass burning
July 2013

kg/m2/day

kg/m2/day

Monthly mean of total column of organic carbon
July 2012

Monthly mean of total column of organic carbon
July 2013

kg/m2

kg/m2

**Figure 9.** Maps of organic carbon emission from biomass burning (top) and total column of primary organic carbon aerosols (bottom) for July 2012 (left) and July 2013 (Right).

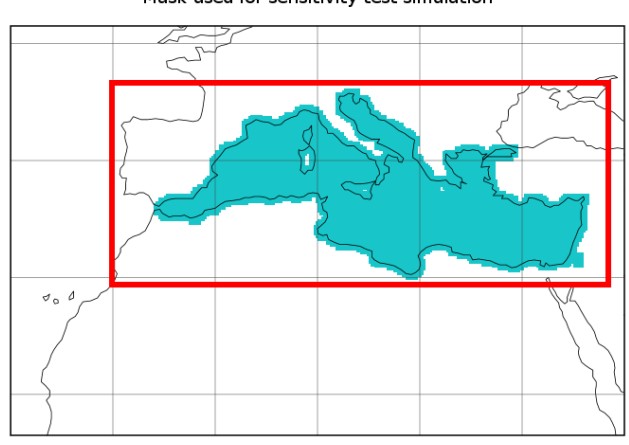

Mask used for sensitivity test simulation

**Figure 10.** Map of the mask used to cancel the anthropogenic emission in the sensitivity test simulation in cyan and the budget domain in red.

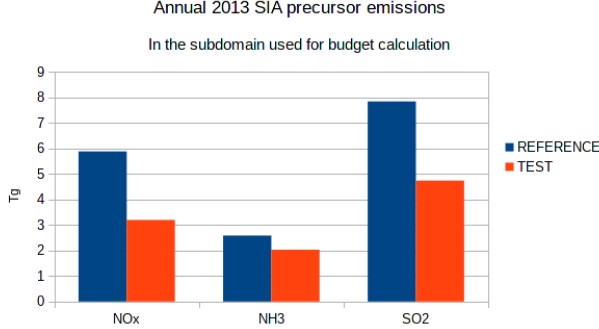

**Figure 11.** Annual emissions of SIA precursors, $NO_x$, $NH_3$ and $SO_2$ for the year 2013 computed over the budget domain in the reference simulation in blue and the sensitivity test in red.

**Table 1.** Mean statistics of the comparison between the MODIS AOD and the MOCAGE simulation for the years 2012 and 2013. See Appendix A for information about statistical indicators.

| Year | Bias | MNMB | FGE | Correlation |
|------|------|------|-----|-------------|
| | | MODIS | | |
| 2012 | −0.04 | −0.09 | 0.39 | 0.39 |
| 2013 | −0.06 | −0.23 | 0.43 | 0.57 |
| | | Deep Blue | | |
| 2012 | −0.13 | −0.37 | 0.61 | 0.39 |
| 2013 | −0.15 | −0.48 | 0.68 | 0.46 |

**Table 2.** Mean statistics of the comparison between the MODIS AOD and the MOCAGE simulation for the years 2012 and 2013 over the budget domain.

| Year | Bias | MNMB | FGE | Correlation |
|------|------|------|-----|-------------|
| | | MODIS | | |
| 2012 | −0.03 | −0.09 | 0.37 | 0.39 |
| 2013 | −0.05 | −0.22 | 0.40 | 0.51 |
| | | Deep Blue | | |
| 2012 | −0.08 | −0.18 | 0.51 | 0.34 |
| 2013 | −0.09 | −0.28 | 0.58 | 0.40 |

**Table 3.** Mean statistics of the comparison between the MODIS AOD and the MOCAGE simulation for the different seasons included in the years 2012 and 2013. See Appendix A for information about statistical indicators.

| Year | Bias | MNMB | FGE | Correlation |
|---|---|---|---|---|
| MODIS | | | | |
| MAM 2012 | −0.05 | −0.13 | 0.40 | 0.34 |
| JJA 2012 | −0.05 | −0.08 | 0.40 | 0.41 |
| SON 2012 | −0.03 | 0.01 | 0.38 | 0.53 |
| DJF 2012/2013 | −0.06 | −0.24 | 0.44 | 0.23 |
| MAM 2013 | −0.06 | −0.23 | 0.42 | 0.35 |
| JJA 2013 | −0.06 | −0.20 | 0.41 | 0.52 |
| SON 2013 | −0.05 | −0.17 | 0.39 | 0.45 |
| Deep Blue | | | | |
| MAM 2012 | −0.13 | −0.33 | 0.57 | 0.37 |
| JJA 2012 | −0.17 | −0.35 | 0.59 | 0.30 |
| SON 2012 | −0.13 | −0.36 | 0.63 | 0.46 |
| DJF 2012/2013 | −0.12 | −0.57 | 0.74 | 0.30 |
| MAM 2013 | −0.14 | −0.44 | 0.66 | 0.40 |
| JJA 2013 | −0.19 | −0.44 | 0.66 | 0.36 |
| SON 2013 | −0.14 | −0.50 | 0.69 | 0.44 |

**Table 4.** Mean statistics of the comparison between the AERONET AOD data and the MOCAGE simulation for the years 2012 and 2013. See Appendix A for information about statistical indicators.

| Year | Bias | MNMB | FGE | Correlation |
|---|---|---|---|---|
| 2012 | −0.01 | 0.10 | 0.41 | 0.69 |
| 2013 | −0.02 | 0.02 | 0.40 | 0.67 |

**Table 5.** Statistics of the comparison between the MOCAGE simulation and the AQeR database measurements, corresponding to classes 1 to 5 from the classification of Joly and Peuch (2012) for the years 2012 and 2013. The table presents results for $PM_{10}$ and $PM_{2.5}$.

| Year | stations | Bias ($\mu gm^{-3}$) | MNMB | FGE | Correlation |
|------|----------|----------------------|------|-----|-------------|
| | | $PM_{10}$ | | | |
| 2012 | 334 | $-9.34$ | $-0.64$ | 0.68 | 0.62 |
| 2013 | 311 | $-8.94$ | $-0.58$ | 0.64 | 0.59 |
| | | $PM_{2.5}$ | | | |
| 2012 | 95 | $-3.49$ | $-0.19$ | 0.54 | 0.71 |
| 2013 | 111 | $-4.13$ | $-0.27$ | 0.54 | 0.68 |

**Table 6.** Statistics of the comparison between the MOCAGE simulation and the AQeR database measurements, corresponding to classes 1 to 5 from the classification of Joly and Peuch (2012) for the years 2012 and 2013 on the budget domain. The table presents results for $PM_{10}$ and $PM_{2.5}$.

| Year | stations | Bias ($\mu gm^{-3}$) | MNMB | FGE | Correlation |
|------|----------|----------------------|------|-----|-------------|
| | | $PM_{10}$ | | | |
| 2012 | 75 | $-10.5$ | $-0.75$ | 0.77 | 0.49 |
| 2013 | 82 | $-10.0$ | $-0.73$ | 0.76 | 0.37 |
| | | $PM_{2.5}$ | | | |
| 2012 | 24 | $-3.91$ | $-0.54$ | 0.71 | 0.45 |
| 2013 | 29 | $-3.51$ | $-0.45$ | 0.65 | 0.48 |

**Table 7.** Statistics of the comparison between the MOCAGE simulation and the AQeR database measurements, corresponding to classes 1 to 5 from the classification of Joly and Peuch (2012) for the different season included in the years 2012 and 2013. The table presents results for $PM_{10}$ and $PM_{2.5}$.

| Year | Bias ($\mu gm^{-3}$) | MNMB | FGE | Correlation |
|---|---|---|---|---|
| | $PM_{10}$ | | | |
| MAM 2012 | $-11.8$ | $-0.75$ | 0.78 | 0.53 |
| JJA 2012 | $-9.61$ | $-0.85$ | 0.85 | 0.44 |
| SON 2012 | $-8.49$ | $-0.62$ | 0.65 | 0.61 |
| DJF 2012/2013 | $-8.63$ | $-0.49$ | 0.55 | 0.67 |
| MAM 2013 | $-11.0$ | $-0.66$ | 0.69 | 0.59 |
| JJA 2013 | $-10.0$ | $-0.78$ | 0.79 | 0.35 |
| SON 2013 | $-6.85$ | $-0.52$ | 0.58 | 0.59 |
| | $PM_{2.5}$ | | | |
| MAM 2012 | $-6.95$ | $-0.58$ | 0.64 | 0.64 |
| JJA 2012 | $-4.28$ | $-0.57$ | 0.63 | 0.46 |
| SON 2012 | $-3.92$ | $-0.35$ | 0.52 | 0.67 |
| DJF 2012/2013 | $-5.18$ | $-0.36$ | 0.54 | 0.71 |
| MAM 2013 | $-6.62$ | $-0.48$ | 0.59 | 0.64 |
| JJA 2013 | $-4.43$ | $-0.47$ | 0.58 | 0.44 |
| SON 2013 | $-2.69$ | $-0.25$ | 0.48 | 0.67 |

**Table 8.** Statistics of the comparison between the MOCAGE simulation and the EMEP measurement database for the year 2013.

| stations | Bias ($\mu gm^{-3}$) | MNMB | FGE | Correlation |
|---|---|---|---|---|
| | Sulphate total | | | |
| 31 | −0.11 | −0.34 | 0.71 | 0.58 |
| | Nitrate | | | |
| 23 | −0.17 | −0.19 | 0.85 | 0.49 |
| | Ammonium | | | |
| 14 | −0.19 | −0.21 | 0.72 | 0.53 |
| | Black carbon | | | |
| 7 | 0.59 | 0.79 | 1.02 | 0.66 |
| | Organic carbon | | | |
| 7 | −2.05 | −0.55 | 0.75 | 0.66 |
| | Sodium | | | |
| 8 | 0.56 | 0.31 | 0.89 | 0.47 |

**Table 9.** Statistics of the comparison between the MOCAGE simulation and the EMEP wet deposition measurements database for the year 2013.

| stations | Bias | MNMB | FGE | Correlation |
|---|---|---|---|---|
| | Sulphate total (Bias in $mgSl^{-1}$) | | | |
| 44 | 0.14 | −0.57 | 1.40 | −0.17 |
| | Nitrate (Bias in $mgNl-1$) | | | |
| 44 | 0.33 | −0.42 | 1.39 | −0.2 |
| | Ammonium (Bias in $mgNl^{-1}$) | | | |
| 44 | 0.77 | −0.28 | 1.40 | −0.15 |
| | Sodium (Bias in $mgl^{-1}$) | | | |
| 44 | 8.90 | 0.17 | 1.48 | 0.05 |

**Table 10.** Annual budget of the total mass of aerosols for the year 2012. The residual mass term corresponds to the values obtained when closing the budget. The different components are in Tg except the mean burden which is in Gg.

| Year 2012 | Emission or Chemical Production | Sedimentation and Dry Deposition | Wet Deposition | Chemical Loss | Import (> 0) Export (< 0) | Mean Burden |
|---|---|---|---|---|---|---|
| Primary Org. C | 1.17 | −0.39 | −0.45 | 0.00 | −0.32 | 34.79 |
| Black carbon | 0.93 | −0.27 | −0.29 | 0.00 | −0.36 | 13.94 |
| Desert dust | 593.6 | −469.7 | −39.9 | 0.00 | −64.1 | 828.9 |
| Sea salt | 298.9 | −273.0 | −17.4 | 0.00 | 2.11 | 146.2 |
| Ammonium | 10.26 | −0.35 | −0.73 | −8.23 | −0.93 | 31.76 |
| Nitrate | 29.04 | −4.89 | −1.66 | −21.8 | −0.62 | 75.65 |
| Sulphate | 7.56 | −3.65 | −1.50 | 0.00 | −2.34 | 69.85 |

**Table 11.** Same as Table 10 but for 2013.

| Year 2013 | Emission or Chemical Production | Sedimentation and Dry Deposition | Wet Deposition | chemical Loss | Import (> 0) Export (< 0) | Mean Burden |
|---|---|---|---|---|---|---|
| Primary Org. C | 1.07 | −0.39 | −0.44 | 0.00 | −0.24 | 33.10 |
| Black carbon | 0.92 | −0.26 | −0.28 | 0.00 | −0.36 | 12.98 |
| Desert dust | 763.5 | −625.3 | −28.9 | 0.00 | −82.1 | 990.2 |
| Sea salt | 309.8 | −282.3 | −19.9 | 0.00 | 6.23 | 152.0 |
| Ammonium | 10.39 | −0.34 | −0.73 | −8.41 | −0.89 | 30.10 |
| Nitrate | 28.48 | −4.76 | −1.66 | −21.4 | −0.61 | 73.38 |
| Sulphate | 7.14 | −3.36 | −1.39 | 0.00 | −2.30 | 65.20 |

**Table 12.** Annual budget of the total mass of aerosols for the year 2012 as a percentage of the emission or the production.

| Year 2012 | Emission or Chemical Production | Sedimentation and Dry Deposition | Wet Deposition | Chemical Loss | Import ($> 0$) Export ($< 0$) |
|---|---|---|---|---|---|
| Primary Org. C | 100% | −33.3% | −38.0% | 0.0% | −27.4% |
| Black carbon | 100% | −29.0% | −31.5% | 0.0% | −38.1% |
| Desert dust | 100% | −79.1% | −6.7% | 0.0% | −10.8% |
| Sea salt | 100% | −91.4% | −5.8% | 0.0% | 0.71% |
| Ammonium | 100% | −3.5% | −7.1% | −80.2% | −9.1% |
| Nitrate | 100% | −16.8% | −5.7% | −75.1% | −2.14% |
| Sulphate | 100% | −48.3% | −19.8% | 0.0% | −34.4% |

**Table 13.** Same as Table 12 but for 2013.

| Year 2013 | Emission or Chemical Production | Sedimentation and Dry Deposition | Wet Deposition | Chemical Loss | Import ($> 0$) Export ($< 0$) |
|---|---|---|---|---|---|
| Primary Org. C | 100% | −36.2% | −40.8% | 0.0% | −22.8% |
| Black carbon | 100% | −28.6% | −30.8% | 0.0% | −39.5% |
| Desert dust | 100% | −81.9% | −3.8% | 0.0% | −10.8% |
| Sea salt | 100% | −91.1% | −6.4% | 0.0% | 2.0% |
| Ammonium | 100% | −3.3% | −7.1% | −80.9% | −8.6% |
| Nitrate | 100% | −16.7% | −5.8% | −75.3% | −2.2% |
| Sulphate | 100% | −47.1% | −19.4% | 0.0% | −35.2% |

**Table 14.** Annual budget for the year 2012 of the total mass of aerosols for the sensitivity test simulation. The differents components are in Tg except the mean burden which is in Gg.

| Year 2012 | Emission or Chemical Production | Sedimentation and Dry Deposition | Wet Deposition | Chemical Loss | Import ($> 0$) Export ($< 0$) | Mean Burden |
|---|---|---|---|---|---|---|
| Primary Org. C | 0.86 | −0.31 | −0.38 | 0.0 | −0.17 | 32.19 |
| Black carbon | 0.66 | −0.20 | −0.23 | 0.00 | −0.22 | 11.55 |
| Ammonium | 7.88 | −0.25 | −0.58 | −6.63 | −0.43 | 26.70 |
| Nitrate | 21.18 | −3.53 | −1.35 | −16.22 | −0.03 | 66.35 |
| Sulphate | 4.28 | −2.67 | −1.14 | 0.00 | −1.04 | 58.03 |

**Table 15.** Same as Table 14 but for 2013.

| Year 2013 | Emission or Chemical Production | Sedimentation and Dry Deposition | Wet Deposition | chemical Loss | Import ($>0$) Export ($<0$) | Mean Burden |
|---|---|---|---|---|---|---|
| Primary Org. C | $-0.76$ | $-0.31$ | $-0.37$ | $0.00$ | $-0.10$ | $30.62$ |
| Black carbon | $-0.64$ | $-0.20$ | $-0.22$ | $0.00$ | $-0.24$ | $10.69$ |
| Ammonium | $-7.68$ | $-0.24$ | $-0.58$ | $-6.76$ | $-0.43$ | $25.38$ |
| Nitrate | $-20.90$ | $-3.48$ | $-1.34$ | $-16.05$ | $-0.04$ | $64.45$ |
| Sulphate | $-4.54$ | $-2.42$ | $-1.05$ | $0.00$ | $-1.11$ | $54.26$ |

**Table 16.** Annual budget for the year 2012 of the total mass of aerosols. The terms corresponds to the relative difference between the reference simulation and the sensitivity test. A negative value means the value is smaller in the test simulation.

| Year 2012 | Emission or Chemical Production | Sedimentation and Dry Deposition | Wet Deposition | Chemical Loss | Import Export | Mean Burden |
|---|---|---|---|---|---|---|
| Primary Org. C | $-26.5\%$ | $-21.4\%$ | $-15.2$ | $N/A$ | $-47.2\%$ | $-7.5\%$ |
| Black carbon | $-29.9\%$ | $-24.6\%$ | $-21.4\%$ | $N/A$ | $-39.6\%$ | $-17.2\%$ |
| Ammonium | $-23.3\%$ | $-29.0\%$ | $-21.0\%$ | $-19.4\%$ | $-54.2\%$ | $-15.9\%$ |
| Nitrate | $-27.1\%$ | $-27.9\%$ | $-18.6\%$ | $-25.6\%$ | $-95.7\%$ | $-12.3\%$ |
| Sulphate | $-35.8\%$ | $-26.9\%$ | $-23.8\%$ | $N/A$ | $-55.2\%$ | $-17.0\%$ |

**Table 17.** Same as Table 16 but for 2013.

| Year 2013 | Emission or Chemical Production | Sedimentation and Dry Deposition | Wet Deposition | Chemical Loss | Import Export | Mean Burden |
|---|---|---|---|---|---|---|
| Primary Org. C | $-28.9\%$ | $-21.5\%$ | $-15.0\%$ | $N/A$ | $-60.0\%$ | $-7.5\%$ |
| Black carbon | $-30.3\%$ | $-25.4\%$ | $-21.6\%$ | $N/A$ | $-34.9\%$ | $-17.6\%$ |
| Ammonium | $-23.2\%$ | $-28.6\%$ | $-21.2\%$ | $-19.6\%$ | $-51.9\%$ | $-15.7\%$ |
| Nitrate | $-26.7\%$ | $-26.8\%$ | $-19.3\%$ | $-25.1\%$ | $-94.2\%$ | $-12.2\%$ |
| Sulphate | $-36.4\%$ | $-27.9\%$ | $-24.0\%$ | $N/A$ | $-51.8\%$ | $-16.8\%$ |