# Peer review of "Primary aerosol and secondary inorganic aerosol budget over the Mediterranean basin during 2012 and 2013"

_Atmospheric Chemistry and Physics, 2017_

## Referee Comment (RC1) · Anonymous Referee #1 · 3 Oct 2017

The paper aims to estimate the aerosol budget over the Mediterranean region using a chemistry-transport model, focusing on two years based on available field campaigns in the Mediterranean region. The model is evaluated against satellite and surface observations. The paper is well-written and easy to follow, with a satisfactory level of the English language. I find the paper publishable in ACP, given the minor comments provided below are addressed.

Introduction

Some key findings of the observations from the three campaigns can be provided in this section. Page 3, lines 3-4: this sentence can be moved to materials and methods

or results, it is not relevant in this part of the introduction.

Mocage model

Can the authors explain the How the vertical and temporal distributions of emissions are done? Does MOCAGE run on hourly resolution input? How about the speciation of NMVOCs in the model?

Evaluation

Please use the full names (e.g. MNMB and other similar acronyms) when they first appear in the text.

What are the units in Tables 1 and 2? Is MNMB in %? Use these units in the text too (e.g. MNMB is lower, between -0.5% and -1%).

Figure 2 shows that AODs are generally underestimated in the eastern Mediterranean in 2013, compared to 2012. Can you comment on this? Is this the meteorology?

Figure 3 can be moved to materials and methods as it does not present any results but the information about the stations and what they measure.

The first 3 paragraphs of section 3.3. also fits better to materials and methods section as it introduces the data used for evaluation.

Is it more correct to say that class 10 represents urban than highly polluted?

The bias in AOD is much smaller compared to bias for the surface stations. Can the authors discuss the reasons for this?

Can the authors explain why they only use EMEP stations?

Some evaluation on deposition using e.g. EMEP stations could be useful as the budget is dependent on this term.

Figure 10 fits better to Materials and Methods.

[Figure]

---

## Referee Comment (RC2) · Anonymous Referee #2 · 23 Oct 2017

This study is a potentially interesting work on the aerosol budget over the Mediterranean during the years 2012 and 2013 but as mentioned by the other reviewer also, the observations from the campaign are not used for the model evaluation, so the only links to the ChArMEx experiment are finally the simulation period and the study area.

The manuscript is well structured but the use of English requires some polishing to be at the level of ACP publications. It could be suitable for publication in ACP after such polishing and a number of further improvements as outlined below.

ÎŹ miss citations in the introduction or comparison of the model results with earlier studies that have shown aerosol budgets in the area like for instance the following

[Figure]

modeling studies:

an de Brugh, J. M. J., Schaap, M., Vignati, E., Dentener, F., Kahnert, M., Sofiev, M., Huijnen, V., and Krol, M. C.: The European aerosol budget in 2006, Atmos. Chem. Phys., 11, 1117-1139, https://doi.org/10.5194/acp-11-1117-2011, 2011.

Im U., Daskalakis N., Markakis K., Vrekoussis M., Hjorth J., Myriokefalitakis S., Gerasopoulos E., Kouvarakis G., Richter A., Burrows J., Pozzoli L., Unal A., Kindap T., Kanakidou M., Simulated Air Quality and Pollutant Budgets over Europe in 2008, Science of Total Environment, 10.1016/j.scitotenv.2013.09.090, 470–471, 270–281, 2014.

Im U., S. Christodoulaki, K. Violaki, P. Zarbas, M. Kocak, N. Daskalakis, N. Mihalopoulos and M. Kanakidou, Atmospheric deposition of nitrogen and sulfur over Europe with focus on the Mediterranean and the Black Sea, Atmospheric Environment, 81, 660-670, http://dx.doi.org/10.1016/j.atmosenv.2013.09.048, 2013

Misisng also reference/comparison to experimental studies like the following data analysis studies that discuss the importance of dust aerosol for the background aerosol in the Mediterranean at surface: Querol, X., Pey, J., Pandolfi, M., Alastuey, A., Cusack, M., Pérez, N., Moreno, T., Viana, M., Mihalopoulos, N., Kallos, G. $\kappa\alpha\iota$ Kleanthous, S.: African dust contributions to mean ambient PM10 masslevels across the Mediterranean Basin, Atmos. Environ., 43(28), 4266–4277, doi:10.1016/j.atmosenv.2009.06.013, 2009.

Querol, X., Alastuey, a., Pey, J., Cusack, M., Pérez, N., Mihalopoulos, N., Theodosi, C., Gerasopoulos, E., Kubilay, N. and Koçak, M.: Variability in regional background aerosols within the Mediterranean, Atmos. Chem. Phys. Discuss., 9(2), 10153–10192, doi:10.5194/acpd-9-10153-2009, 2009.

There are also studies that investigate the entire tropospheric column over the Mediterranean, based both on ground based observations and satellite observations.

How the results of the present study compare with these earlier studies?

Also in all tables and throughout the manuscript where reference is made to budget, it has to be clarified that they concern PM10 aerosol (or is it bulk aerosol in the model?).

Specific comments:

Abstract: line 6: annual AEROSOL burden

Section 2.1 MOCAGE model: Information is missing on how the aerosol size is parameterized in the model. Since later in the discussion the authors refer to the aerosol size to explain the model results, it is appropriate to briefly outline what aerosol sizes are taken into account in the model and how they change during atmospheric aging.

Section 2.1 last paragraph – here the authors could refer to observationally based studies (Querol et al) the estimates the importance of dust aerosol for the background aerosol levels in the Mediterranean.

Section 3.1

In the comparisons with the databases (MODIS, AERONET, AIRBASE / AQeR, EMEP), a seasonal comparison would be more useful than an annual.

lines 14-15: comment on the underestimate of dust aerosol, what are the potential reasons?

line 20 : domains

Line 24 : simulate

Page 6, line 3: numbers

Line 19 classes 1 to 5 are kept in order to remove . . .

Section 3.4: it seems that no data from the Mediterranean region are included; this has to be explicitly mentioned in the text since it is a limitation of the comparison

Page 8, line 5: Δburden remove 'is'

Section 4.2.1 you need to define the exact 'box' in which you calculate the budget both horizontal extent and vertical extent of the area.

Page 9, line 1 in the region;

Page 9, line 2 define SIA

Page 9, line 3: replace provide by add up to

Page 9, line 5: once can also note...

Page 9, 1st paragraph: please explain what are the chemical loss terms for nitrate and ammonium aerosol.

Page 9, line7, add after 'whatever the thermodynamic conditions are' ' because of the very low vapor pressure of the sulfuric acid'. And remove the sentence after that.

Page 9, lines 15-18: Nitrate aerosol partitioning to the aerosol phase is very sensitive to the aerosol pH (Guo, H., et al. (2016), Fine particle pH and the partitioning of nitric acid during winter in the northeastern United States, J. Geophys. Res. Atmos., 121, doi:10.1002/2016JD025311) and NH4NO3 is semi volatile. How well is this computed in your model? Do you consider dust aerosol alkaline components in ISORROPIA calculations?

In Tables 10 and 11, the percentages do not add up to 100% per species (no closure-could a mass balance issue?) and the differences from the 100% seem to be larger in 2012 than in 2013. Why there are differences? Is it a question of spin up time for the model stabilization? Is there a significant change in the burden ? and if yes is such change justified and by what?

Tables 14 and 15 are produced from Tables 8 and 9 (all emissions) when compared to Tables 12 and 13 (no anthropogenic emissions over sea and at the coasts). However, the percentages provided in Table 14 and 15 are similar but not the same as those calculated when using numbers from the relevant Tables. The numbers have to be

double checked and the reason of this difference has to be clarified.

Page 10, lines 7 and 8 replace 'season' by 'dust period'

Page 10, line 10, over North Africa

Figure 9 could be in the supplement since it is a global model result.

Page 10, line 22: annual cycle with maximum burden in summer

Page 10, line 28: from which boarder(s) of the domain the export takes place?

Page 12, 1st paragraph: also nitrate partitioning depends on the aerosol pH (see earlier comment).

Page 12, line 10 – mention also that many major cities in this area are coastal.

Page 6, the values reported for Table 4 are different in the text than in the table.

---

## Referee Comment (RC3) · Anonymous Referee #3 · 25 Oct 2017

Primary aerosols and secondary inorganic aerosols budget over the Mediterranean basin during 2012 and 2013 by Jonathan Guth, Virginie Marécal, Béatrice Josse, and Joaquim Arteta

In this article, the authors present the results of MOCAGE CTM simulations of aerosol concentrations and burdens in the Mediterranean basin for 2012-2013. The model evaluation was performed against Airbase and EMEP surface monitoring data and MODIS and AERONET AOD measurements. The authors present model calculated contribution of different aerosol types to the aerosol columns and surface concentrations and the aerosol budget for a selected area, both on the annual and monthly basis,

focusing the study on primary (anthropogenic and natural) and secondary inorganic particles. An additional test was performed to estimate the importance of Mediterranean shipping emissions (referred to as anthropogenic in the paper) and land-based coastal anthropogenic emissions for the region of the study.

The paper is concerned with an interesting and challenging issue of characterizing the tropospheric aerosol in the Mediterranean region which is influenced by the emissions from a variety of anthropogenic and natural sources. Thus, the subject of the work is quite relevant for better understanding the nature of the aerosol pollution and the role of different sources and processes in the area, and could potentially contribute to design more optimal strategies to improve air quality.

The material is presented relatively clear and supported/illustrated amply (maybe even too many) by the tables and figures, though the manuscript needs more work to check on the grammar and correct awkward formulations.

The results definitely contain interesting assessments, but still I have a big problem to see the purpose of the publication and usefulness of the results with respect to aerosol budgets for a seemingly random rectangle area, and also the purpose of performing the "sensitivity (?) test" for "marine and coastal areas". Perhaps the authors should clarify/justify the aim of the work. As a part of that, try to better explain their choice of the study region (the red rectangle) for the "Mediterranean basin" (this is only mentioned for dust on p. 12, line 22) and also 50 m wide coast line. I'd recommend that the authors make a better effort to show the added value of the obtained results and their scientific and/or practical usefulness and implication.

Other general comments:

1. The authors should make more clear the connection to the ChArMEx and the choice of the years, since they have not at all made any use of the campaign's data.

2. I'd recommend to use (international) shipping or ship emissions instead of anthropogenic emissions from the sea. By the way, would not it be more useful to assess separately the role of international shipping and land-based emission in the aerosol pollution in the region?

3. Please, specify what were aerosol sizes included in the study (p.3 l.30-31), in particular for sea salt and dust (p.4)

4. Why have not you limited your model evaluation with EMEP data to secondary inorganics. Why would not you also evaluate PM10, PM2.5, elemental carbon, Na+ (perhaps even dust)?

5. to p. 9. African dust is often transported at higher levels in BL and free troposphere. Please justify the use of 200m hight winds for describing dust transport here. Have you looked at 3d dust concentrations/vertical profiles?

6. How sensitive the results of the study to the choice of the study area (the location and size of the red rectangle)? See p. 12 line 22

7. Almost half of the Conclusions section is actually Outlook. I'd recommend to improve the Conclusions, avoid general statements and highlight your findings.

8. Decide whether to use "sea salt" or "sea salts"; better to use import/export than importation/ exportation

Minor/editing comments:

L 4-5: why not write like 29°N , 10°W

L 8: this is quite "normal" seasonal variation – not sure it should be in Abstract

L11: "with high values for some of them" - specify or drop out; Maybe better to give a range of values for different aerosols than just BC example (40%).

L 17-18: I'm not sure it's quite true for dust, since you approach doesn't distinguish

between export and import.

L 22: on weather (meteorology is a scientific study)

L 23: What do you mean by "sensitive to atmospheric pollution"? In what terms? What is the difference here between atmospheric pollution and air quality?

Page 2:

L 1-2: experiences periodic/sporadic pollution from forest fires? Complex topography? Check the sentence – geography/topography associated with the flows? Are you sure/What do mean by "it's especially sensitive to climate change"?

L 7: in summer months in 2012 and 2013

L 13-14: strange sentence

L 9-19: not sure these details about weather during those short periods are relevant, but more information about "data collected" could be.

L 20-24: aerosol contribution to what? Edit: composed of; were the second large contributor. anthropogenic emissions were the major part of PM2.5 composition.

L 25-26: re-write the sentence

L 28: aerosols.. were dominated by. . .

L 32: ..two years .. include the the intensive periods. . .

L 15: the importance of SOA - inconsistent with p. 3 (31032)

L 22 (also p.6 l.3): Are you discussing here MOCAGE vs CHIMERE? Is it in the paper's scope?

L 31: for 2012 and 2013

L 8-14: not sure it's needed if just only the data format was changed, unless there were more essential changes affecting the data consistency

L 15: periurban?

L 16: for model evaluation..;

L 18-19: rural station; 1 to 5 are kept in order to assure a set of representative sites (or whereas 6 to 10 that are not representative are removed)

L 22: "similar behaviour" - awkward formulation; aerosols should be PM10 and PM2.5

L 24: PM2.5 is better. . .

L 26: you mean that the natural aerosols are accurately calculated?

L 28-29: "slightly less good" = worse

L 1-5: If you used EMEP monitoring data (from ebas.no data base), you should write that explicitly. There is no need to list US, Asian etc networks/databases.

L 6: ..use measurements of secondary inorganic aerosols. . .

L 23: can you specify the errors?

L 8 and 14: Inconsistency: All terms...directly computable, but indirect estimation Ttran ???

L 3: primary carbonaceous

L 5-7: since you are discussing the hypothesis, how sound/unsound you mean it its,

how much uncertainty in the results it causes?

L 15: What about the export from the Atlantic? Could it be that import/export are considerable, but compensate each other in the budget?

L 19: to further facilitate the analysis. . ..

L 22: "north of the southern boundary" sounds strange, maybe along the southern boundary of the domain;

L 23: I do not understand how E/N-E winds can transport dust from the south.

L 24-25: for carbonaceous aerosols; across the eastern border

L 26: Explain that you compare the precipitation amount and wind speeds in 2012 and 2013

L 30: sea salt aerosol levels

L 31: less precipitation, in exactly which area – impossible to tell from Fig. 6

l 33: higher speeds of the wind advecting/bringing transporting the pollution

L 2-3: the monthly variations of budgets, better use the Figure present or show

l 5-6: may be to talk about larger and smaller emissions, or higher and lower dust levels instead of active/less active season

L 10. winds were stronger in North Africa. . .

L 14: not in summer, but in July month

L 19: likely reason for the enhanced BC concentrations in July 2013

L 26: bad formulation. Suggested: there are considerable differences in the tropospheric (?) load of the different aerosols.

L 29-31: does not the same (emissions and meteorological conditions within the domain) applies to sea salt; also explain "meteorological conditions within the domain"

L 33: Re-write sentence starting with Another source...

L 17. The model does not equally "well represents the aerosols". Please, make more precise summary. L 21: ..all considered in the study aerosols are exported

L 22-25: Re-write awkward sentences: We observed an annual cycle... (what's new about this?) and The annual cycle can also be affected.. ( has to be more specific; what difference in primary emissions?...)

L 31: Re-write the last sentences

Fig 2: make the circles visible

Figs 4-5: difficult to see the differences. Maybe to plot the difference instead?

Fig 7: correct caption: sea salt

Table 7 caption: comparison MOCAGE simulations with EMEP observations

---

## Author Comment (AC1) · 21 Dec 2017

**Response to anonymous Referee number 1 review**

The authors would like to thank referee #1 for his/her useful comments. Each response to the referee's question is organized as follows: (1) comment from the referee in bold, (2) authors' response and changes in the manuscript in normal font. The changes in the revised manuscript, except the small edit corrections, are in green color in the revised manuscript. Moreover, the manuscript has been proofread by a native English speaker.

1. **Some key findings of the observations from the three campaigns can be provided in this section.**

   It is a good suggestion. Nevertheless, the different published results mainly concern radiative aspects, secondary organic aerosols or look at very small scale. This is then difficult to fit key findings into the scope of our study to add in the introduction. However, we compared our results to pertinent studies into the main part on the manuscript. In addition, the introduction has been completed with details about measurements made during the ChArMEx related campaigns.

2. **Page 3, line 3-4; this sentence can be removed to materials and methods or results, it is not relevant in this part of the introduction**

   The sentence has been removed.

3. **Can the author explain how the vertical and temporal distributions of emissions are done? Does MOCAGE run on hourly resolution input? How about the speciations of NMVOCs in the model?**

   In the model, all emissions are distributed on the 5 lowest model layers using an exponential decay with a decay constant of 5. The temporal distribution is based on the monthly variations provided in the inventory chosen on top of which are added the variations linked to the day of the week and the time of the day (following EMEP profiles). The NNMVOCs speciation is based following Stockwell et al., (1997) based on Middleton et al, (1990). This information has been added to the manuscript.

4. **Please use the full names (e.g. MNMB and other similar acronyms) when they first appear in the text**

   The acronyms for MNMB and FGE have been added to the text when they first appears.

5. **What are the units in Tables 1 and 2? Is MNMB in $\%$ ? Use these units in the text too (e.g. MNMB is lower, between $-0.5\%$ and $-1\%$).**

   AOD being a dimensionless quantity, the bias is also dimensionless. MNMB and FGE are always dimensionless measures. This has been added in the text.

6. **Figure 2 shows that AODs are generally underestimated in the eastern Mediterranean in 2013 compared to 2012. Can you comment on this? is it the meteorology?**

   Emission inventories are not well known in this area and likely underestimated leading to a systematic negative bias of the model over this region. The year 2012 has more rainfall in this area, leading to more wet deposition (Fig. 6). The mean concentrations in 2012, and also the AOD, is then lower and closer in the simulation to the reality since wet deposition reduces the impact of emission uncertainty. This has been added in the text.

7. **Figure 3 can be moved to materials and methods as it does not present any results but the information about the stations and what they measure. The first 3 paragraphs of section 3.3 also fits better to materials and methods section as in introduces the data used for evaluation.**

   We think that Figure 3 can be kept in the main part of the manuscript since the reader can easily refer to it when reading the associated comment. For the first 3 paragraphs of section 3.3, we agree that they are not essential in the analysis of the results. They have been moved to the appendix.

8. **"Is it more correct to say that class 10 represents urban than highly polluted?"**

   The classification method uses the temporal variation of the time-series at each station to determine its class and the class 10 corresponds to stations characterized as traffic. The text has been changed.

9. **The bias in AOD is much smaller compared to bias for the surface stations. Can the authors discuss the reason for this?**

   The AOD is a measure of the integrated column of aerosols while surface stations aim at evaluating the aerosol concentrations at one specific level. There can be a good agreement with AODs while not with surface stations if the vertical distribution is not correct and/or if the particles are not well distributed in size.

10. **Can the authors explain why they only use EMEP Stations**

    There is a problem with the words employed. What we wanted to say is that the choice of the domain makes that only EMEP stations are located into the simulation domain. It has been corrected in the text.

11. **Some evaluation on deposition using e.g. EMEP stations could be useful as the budget is dependent on this term**

    Following your advice, we added a comparison to wet deposition data from the EMEP stations to the manuscript.

12. **Figure 10 fits better to Materials and method**

    We think that Figure 10 can be kept in the main part of the manuscript since the reader can easily refer to it when reading the analysis of the results of the test experiment.

---

## Author Comment (AC2) · 21 Dec 2017

**Response to anonymous Referee number 2 review**

The authors would like to thank referee #2 for his/her useful comments. Each response to the referee's question is organized as follows: (1) comment from the referee in bold, (2) authors' response and changes in the manuscript in normal font. The changes in the revised manuscript, except the small edit corrections, are in green color in the revised manuscript. Moreover, the manuscript has been proofread by a native English speaker.

1. **How the results of the present study compare with these earlier studies?**

   We added comparisons to previous studies at several places in the text. In particular, we compared the results of our budget calculation to previous budget over Europe and found generally good consistency, such as the fact that the zone in a net source of pollutants as already shown by Aan de Brugh et al., (2011) and Im et al., (2014).

2. **Also in all tables and throughout the manuscript where reference is made to budget, it has to be clarified they concerne PM10 aerosol (or is it bulk aerosol in the model?)**

   All the budgets presented in the manuscript are for total aerosol mass, not PM10. This has been clarified in section 4.2.

3. **Abstract line 6: annual AEROSOL burden**

   Added to the manuscript.

4. **Section 2.1 MOCAGE model Information is missing on how the aerosol size is parametrized in the model. Since later in the discussion the authors refer to the aerosol sizes are taken into account in the model and how they change during atmospheric aging.**

   The model uses a sectional representation with six size bins for each aerosol type. The sizes are ranging from to 2nm to 50μm and there is no effect of ageing on aerosol size in the model. We added this information and also added the size parametrization used for the desert dust that was missing in the manuscript (Kok, 2011).

5. **Section 2.1 last paragraph - here the authors could refer to observationnally based studies (Querol et al) the estimates the importance of dust aerosol for the background aerosol levels in the Mediterranean.**

   The reference to the work of Querol at al., (2009) has been added to the text.

6. **Section 3.1. In the comparison with the database (MODIS, AERONET, AIRBASE/AQeR, EMEP), a seasonal comparison would be more useful than an annual.**

   Following your remark, we added, when it was informative (MODIS and AQeR) a seasonal comparison to the evaluation of the simulation in the manuscript.

7. **line 14-15: comment on the underestimate of dust aerosol, what are the potential reasons?**

   A potential explanation maybe that the desert dust aerosols are not transported far enough, the vertical injection and the following transport being hard to model properly.

8. **line 20: domains**

   Corrected.

9. **line 24: simulate**

   Corrected.

10. **Page 6, line 3: numbers**

    Corrected.

11. **line 19 classes 1 to 5 are kept in order to remove ...**

   Corrected.

12. **Section 3.4: it seems that no data from the Mediterranean region are included: this has to be explicitly mentioned in the text since it is a limitation of the comparison**

   The text mention the "lack of this type of measurements outside Europe" highlighting the limitation of the comparison. To be more precise, we added a sentence saying that EMEP network only allows us only to characterize the north west part on the Mediterranean and that this is a limitation of the comparison.

13. **Page 8, line 5: $\Delta$burden remove 'is'**

   Corrected.

14. **Section 4.2.1 you need to define the exact 'box' in which you calculate the budget both horizontal and vertical extent of the area** Done in section 4.2

15. **Page 9, line 1 in the region;**

   Corrected.

16. **Page 9, line 2 Define SIA**

   Added to the manuscript.

17. **Page 9, line 3: Replace provide by add up to**

   Corrected.

18. **Page 9, line 5: One can also note...**

   Corrected.

19. **Page 9, 1st paragraph, please explain what are the chemical loss terms for nitrate and ammonium aerosol.**

   For ammonium, nitrate and sulphate there are no emissions in the model. The first columns corresponds to the quantity of aerosol condensed. In the same way, the column "chemical loss" shows the evaporation of the aerosols. This has been added to the manuscript.

20. **Page 9, line 7, add after 'whatever the thermodynamic conditions are' 'because of the very low vapor presure of the sulfuric acid'. And remove the sentence after that:**

   Done.

21. **Page 9, line 15-18: Nitrate aerosol partitioning to the aerosol phase is very sensitive to the aerosol pH (Guo et al., 2016) and NH4NO3 is semi volatile. How well is this computed in your model? Do you consider dust aerosol alkaline components in ISORROPIA calculations?**

   The partitioning with the pH is treated by ISORROPIA (we are using version 2 of ISORROPIA), the input being the the ions concentrations and the water content. Although the pH has not been validated extensively in our model, it has been shown by Guo at al, (2015) that ISORROPIA II predicts well the pH. The water content input is provided by the numerical weather prediction simulation, in which data assimilation is made. Therefore, we assume the input information is as precise as possible. Concerning, the alkaline components are not yet considered in this study. A work is in progress to include it into the ISORROPIA implementation in MOCAGE. The inclusion of these compounds would increase pH and shift the partitioning of HNO3 to the aerosol phase.

22. **Table 10 and 11, the percentage do not add up to $100\%$ per species (no closure - could a mass balance issue?) and the differences from the $100\%$ seem to be larger in 2012 than in 2013. Why there are differences. is it a question of spin up time for the model stabilization? Is there a significant change in the burden ? and if yes is such change justified and by what?**

The tables in the manuscript were not clear. The reader might have thought, for the first line of table 10, than 33.3 + 38 - 27.4 should be equal to 100. In order to make the manuscript clearer, we changed the signs of the numbers in the tables. All sinks are now negative.

Moreover, the term of transportation is computed as the difference between the mass before and after the advection. Hence this term includes the error made during the advection which remain small but can become not negligible as we are using a semi-lagrangian scheme which has a smoothing effect on the strong peak in aerosols concentrations leading to a small loss of mass. These is discussed in section 4.1. This can be seen for example with desert dust where the residual is the higher. Moreover, the difference between 2012 and 2013 can be explained by the difference in the concentration field to be advected. For example, the desert dust emissions are higher in 2013 where the residual mass is slightly higher too. There is no spin up effect since the two years of simulation are made the same way, with the same 3 months spin-up. This has been added to the manuscript.

23. **Tables 14 and 15 are produced from Tables 8 and 9 (all emissions) when compared to Tables 12 and 13 (no anthropogenic emissions over sea and at the coasts). However,the percentages provided in Table 14 and 15 are similar but not the same as those calculated when using numbers from the relevant Tables. The numbers have to be double checked and the reason of this difference has to be clarified.**

It comes from the fact that the numbers in the tables 14 and 15 are computed with the real values of the budget, not the round numbers. For example for the very first term of the table 15 values in the manuscript are 1.07 and 0.76 which lead to $(0.76-1.07)/1.07*100 = -28.97\%$. But the "real" values are 0.763 and 1.073 which leads to $(0.763-1.073)/1.073 = -28.89\%$. The text has been changed to explain why the reader could find some small differences.

24. **Page 10, lines 7 and 8: replace season by dust period:**

Corrected.

25. **Page 10, line 10: over North Africa:**

Corrected.

26. **Figure 9 could be in the supplement since it is a global model result.**

Since this figure would be the only material in the supplement we prefer keep it in the manuscript.

27. **Page 10, line 22: annual cycle with maximum burden in summer:**

Changed.

28. **Page 10, line 28: from which boarder(s) of the domain the export takes place?**

With our computation method we can not have the exact position where the exportation takes place. Based on concentration and wind maps, we showed for example that desert dust aerosols are exported out of the study domain through the southern boundary, while carbonaceous aerosols are mainly exported out of the study domain through the eastern boundary. This has been added to the text.

29. **Page 12; 1st paragraph: also nitrate partitioning depends on the aerosol pH (see earlier comment)**

We added a sentence in the text to add this argument.

30. **Page 12, line 10 - mention also that many major cities in this area are coastal**

Added.

31. **Page 6, the values reported for Table 4 are different in the text than in the table**

This has been corrected.

---

## Author Comment (AC3) · 21 Dec 2017

**Response to anonymous Referee number 3 review**

The authors would like to thank referee #2 for his/her useful comments. Each response to the referee's question is organized as follows: (1) comment from the referee in bold, (2) authors' response and changes in the manuscript in normal font. The changes in the revised manuscript, except the small edit corrections, are in green color in the revised manuscript. Moreover, the manuscript has been proofread by a native English speaker.

1. **Perhaps the authors should clarify/justify the aim of the work. As a part of that, try to better explain their choice of the study region (the red rectangle) for the "Mediterranean basin" (this is only mentioned for dust on p. 12, line 22) and also 50 m wide coast line. I'd recommend that the authors make a better effort to show the added value of the obtained results and their scientific and/or practical usefulness and implication. The authors should make more clear the connection to the ChArMEx and the choice of the years, since they have not at all made any use of the campaign's data.**

   Some of the points of this remark will be addressed later in this document. The choice of the study domain is addressed in points 2 and 6.

   We agree that the paragraph in the introduction describing the aims of the work was not clear enough and also the link to ChArMEx. The paragraph of the introduction providing the aims of the paper has been changed.

   « The past studies focused on summer season. Here we go a step further by analysing the aerosols over the Mediterranean region based on a two year long simulation that includes the intensive periods (2012 and 2013). Our objective is to establish the budget of the primary aerosols and secondary inorganic aerosols in this region for these two years including an analysis of its seasonal variability. Because particulate pollution is an issue there, we also analyse, from a sensitivity simulation, the contribution of the anthropogenic emissions from in the Mediterranean coast and from international shipping emissions to the aerosol budget. The years 2012 and 2013 having different mean meteorological conditions, this allows us to quantify the impact of this year-to-year meteorological variability on the aerosol budget. This work being part of the ChArMEx project, the choice of the years 2012 and 2013 for this study was linked to the possible availability of ChArMEx data, such as aerosol composition, on the long term that could be compared to the simulation results. Unfortunately, to day, these data are not available. Nevertheless, these years are of interest because of their different meteorology and also it makes possible to link our results to other ChArMEx studies published in the present issue. »

   The conclusion has also been changed to make clearer the objectives of this study.

   «This study aimed at establishing the budget of the primary aerosols and secondary inorganic aerosols on the Mediterranean basin based on numerical simulations of the years 2012 and 2013 using the MOCAGE model. We also studied its seasonal variability, its year-to-year variability with meteorological conditions, and the contribution of local anthropogenic emissions from the populated coastal area and from international shipping in the Mediterranean basin. Firstly, [...] ».

2. **I'd recommend to use (international) shipping or ship emissions instead of anthropogenic emissions from the sea. By the way, would not it be more useful to assess separately the role of international shipping and land-based emission in the aerosol pollution in the region?**

   The Mediterranean basin is a region of high economic activity linked to the shipping business. The development of both coastal land-based and international shipping economy are linked and should not be dissociated according to the authors. The reference to international shipping emissions has been corrected.

3. **Please, specify what were aerosol sizes included in the study (p.3 l.30-31), in particular for sea salt and dust (p.4)**

   The aerosol sizes in the model have been added in the model description.

[Figure]

**Figure 1.** Yearly mean of wind vectors at 1000m above the surface for the year 2013.

4. **Why have not you limited your model evaluation with EMEP data to secondary inorganics. Why would not you also evaluate PM10, PM2.5, elemental carbon, Na+ (perhaps even dust)?**

   The PM10 and PM2.5 data are already evaluated using the AQeR database which has a much larger number of stations than the EMEP database. Following your suggestion, we added the comparison for elemental carbon and organic carbon and sodium in the manuscript.

5. **to p. 9. African dust is often transported at higher levels in BL and free troposphere. Please justify the use of 200m hight winds for describing dust transport here. Have you looked at 3d dust concentrations/vertical profiles?**

   Indeed desert dust can be transported in the free troposphere. The wind map at 200m is representative of the global wind in the low troposphere. Figure 1 present the wind map at 1 km above the surface for the year 2013. The wind vectors in this figure presents the same patterns and structure as at 200m above the surface (Fig. 6 in the manuscript). Moreover the 200m level is closer to the surface, hence it is more representative for the desert dust emissions behaviour.

6. **How sensitive the results of the study to the choice of the study area (the location and size of the red rectangle)? See p. 12 line 22**

   We chose a domain that contained all the basin and the coast to account for the inseparable maritime and harbour activities. The complex pattern of the seaside coast forces us to sometimes take a large part of the inland (e.g. in North Africa). Moreover we wanted a rectangle domain to ease the computation and the analysis regarding meteorological parameters which would have not been possible with such a discontinuous domain. We have not tested the sensibility of the location of the limit of the domain used.

Nevertheless, we expect the sensitivity to vary from one aerosol to another. It is important for desert dust, as we have seen, the southern boarder of the study domain crosses an emission area. For sea salt for example we expect the results to be less sensitive as they are not transported far away due to their large size. For primary organic carbon and black carbon, we also expect the result to be not very sensitive as the export essentially is linked to high concentrations in the Middle East associated with a western flux exporting them.

7. **Almost half of the Conclusions section is actually Outlook. I'd recommend to improve the Conclusions, avoid general statements and highlight your findings.**

The conclusion has been partially rewritten and completed.

8. **Decide whether to use "sea salt" or "sea salts"; better to use import/export than importation/ exportation**

The text has been changed to use sea salt and import/export.

9. **Page 1 L 4-5: why not write like $29°$ N , $10°$ W**

ACP's author guidelines only claim coordinates need a degree sign and a space when naming the direction. It is the author choice to write it this way.

10. **L 8: this is quite "normal" seasonal variation – not sure it should be in Abstract**

The sentence has been removed.

11. **L11: "with high values for some of them" - specify or drop out; Maybe better to give a range of values for different aerosols than just BC example (40%).**

The sentence has been removed.

12. **L 17-18: I'm not sure it's quite true for dust, since you approach doesn't distinguish between export and import.**

Here, the sentence only applies to anthropogenic emissions as written line 15. Since it seemed not clear enough we added it also line 16.

13. **L 22: on weather (meteorology is a scientific study)**

Corrected

14. **L 23: What do you mean by "sensitive to atmospheric pollution"? In what terms? What is the difference here between atmospheric pollution and air quality?**

We agree that we have misused the word sensitive. What we wanted to say is that the Mediterranean basin is subject to atmospheric pollution issues. The difference between atmopsheric pollution and air quality is that air quality is one of the many issues that can be related to atmospheric pollution.

15. **Page 2: L 1-2: experiences periodic/sporadic pollution from forest fires? Complex topography? Check the sentence – geography/topography associated with the flows? Are you sure/What do mean by "it's especially sensitive to climate change"?**

Again, the word "sensitive" was misused. The idea here was to say that the climate simulations tend to show that the climate of the Mediterranean basin will become dryer and warmer, especially during summer.

16. **L 7: in summer months in 2012 and 2013**

Corrected.

17. **L 13-14: strange sentence**

The sentence has been simplified.

18. **L 9-19: not sure these details about weather during those short periods are relevant, but more information about "data collected" could be.**

The introduction has been completed with details about measurements made during the ChArMEx related campaigns.

19. **L 20-24: aerosol contribution to what? Edit: composed of; were the second large contributor. anthropogenic emissions were the major part of PM2.5 composition.**

The sentences have been changed.

20. **L 25-26: re-write the sentence**

Done.

21. **L 28: aerosols.. were dominated by. . .**

Corrected.

22. **L 32: ..two years .. include the the intensive periods. . .**

Corrected

23. **Page 5 L 15: the importance of SOA - inconsistent with p. 3 (31032)**

We do not understand where the inconsistency is. Page 5 L15 claims SOA can "represent a significant part of the fine mode aerosols" while page 3 l 31-32 claims that "the fine mode aerosol contribution [to **total mass**] is low". We added the words "to total mass" in the manuscript to be more precise.

24. **L 22 (also p.6 l.3): Are you discussing here MOCAGE vs CHIMERE? Is it in the paper's scope?**

No we are comparing our results to a recent study over similar domain and period.

25. **L 31: for 2012 and 2013**

Corrected.

26. **Page 6 L 8-14: not sure it's needed if just only the data format was changed, unless there were more essential changes affecting the data consistency**

These sentences present the data used in the study. There is now only one sentence about the change between AIRBASE ans AQeR. The first three paragraphs have been moved to the Appendix following referee 2.

27. **L 15: periurban?**

Corrected to peri-urban (https://www.eea.europa.eu/themes/sustainability-transitions/urban-environment/urban-green-infrastructure/glossary-for-urban-green-infrastructure)

28. **L 16: for model evaluation..;**

Corrected.

29. **L 18-19: rural station; 1 to 5 are kept in order to assure a set of representative sites (or whereas 6 to 10 that are not representative are removed)**

Corrected.

30. **L 22: "similar behaviour" - awkward formulation; aerosols should be PM10 and PM2.5**

In this sentence we talk about the years 2012 and 2013, not PM10 and PM2.5. The sentence has been changed to be clearer.

31. **L 24: PM2.5 is better. . .**

This sentence has been changed.

32. **L 26: you mean that the natural aerosols are accurately calculated?**

We added a sentence to clarify that in the region where measurements of the AQeR database are made natural aerosols do not play an important role.

33. **L 28-29: "slightly less good" = worse**

Corrected.

34. **Page 7 L 1-5: If you used EMEP monitoring data (from ebas.no data base), you should write that explicitly. There is no need to list US, Asian etc networks/databases.**

Corrected.

35. **L 6: ..use measurements of secondary inorganic aerosols. . .**

Modified.

36. **L 23: can you specify the errors?**

The MOCAGE model does not represent SOA, so that we do not have available quantity to estimate the errors.

37. **Page 8 L 8 and 14: Inconsistency: All terms...directly computable, but indirect estimation Ttran ???**

The word "prognostic" has been changed to "estimated" to correct the sentence.

38. **Page 9, L 3: primary carbonaceous**

Corrected

39. **L 5-7: since you are discussing the hypothesis, how sound/unsound you mean it its, how much uncertainty in the results it causes?**

The authors are not discussing the hypothesis made. They are simply explaining why the destruction of sulphate aerosols is equal to zero. According to Fountoukis and Nenes (2007), "Sulfuric acid, sodium and crustal species have a very low vapor pressure and are assumed to exclusively reside in the aerosol phase". The sentence has been changed in the manuscript for a better understanding.

40. **L 15: What about the export from the Atlantic? Could it be that import/export are considerable, but compensate each other in the budget?**

The authors wrote: "We can then consider the global behaviour of sea salt is that there is almost no flux." This means that over the domain considered, the total import/export of sea salt is balanced. We can not give more information with our method used for computing the budget since we cannot differentiate import from export.

41. **L 19: to further facilitate the analysis...**

Corrected.

42. **L 22: "north of the southern boundary" sounds strange, maybe along the southern boundary of the domain;**

Corrected.

43. **L 23: I do not understand how E/N-E winds can transport dust from the south.:**

They transport desert dust, emitted into the study domain, towards the south, outside the domain of study. The sentence has been rewritten to be clearer.

44. **L 24-25: for carbonaceous aerosols; across the eastern border**

Corrected.

45. **L 26: Explain that you compare the precipitation amount and wind speeds in 2012 and 2013**

This was mentioned in line 21, but we added it in line 26 also.

46. **L 30: sea salt aerosol levels**

Changed.

47. **L 31: less precipitation, in exactly which area – impossible to tell from Fig. 6**

Corrected.

48. **l 33: higher speeds of the wind advecting/bringing transporting the pollution**

Corrected.

49. **Page 10 L 2-3: the monthly variations of budgets, better use the Figure present or show**

Corrected.

50. **l 5-6: may be to talk about larger and smaller emissions, or higher and lower dust levels instead of active/less active season**

It has been corrected by talking about dust levels.

51. **L 10. winds were stronger in North Africa. . .**

Corrected.

52. **L 14: not in summer, but in July month**

The sentence has been corrected by saying that the burden is higher or equivalent despite lower emissions.

53. **L 19: likely reason for the enhanced BC concentrations in July 2013**

The authors are not sure about the question asked here. We assume the question is "Why does not BC also have en-hanced concentrations in July 2013?". The enhanced concentrations of OC in July 2013 originate from biomass burning emissions in North America. The emission inventory used gives much more OC emissions than BC for these events, explaining the difference of behaviour.

54. **L 26: bad formulation. Suggested: there are considerable differences in the tropospheric (?) load of the different aerosols.**

Corrected.

55. **L 29-31: does not the same (emissions and meteorological conditions within the domain) applies to sea salt; also explain "meteorological conditions within the domain"**

Since the emissions are dependent of the 10m wind speed, they have spatial variations according to the spatial pattern of the wind field. Hence, they are not the same inside and outside the domain of study. The "meteorological conditions within the domain" are the metorological conditions inside the domain. This has been changed in the manuscript.

56. **L 33: Re-write sentence starting with Another source...**

Done.

57. **Page 12 L 17. The model does not equally "well represents the aerosols". Please, make more precise summary.**

This part of the conclusion has been rewritten considering the changes made in the manuscript.

58. **L 21: ..all considered in the study aerosols are exported**

Changed.

59. **L 22-25: Re-write awkward sentences: We observed an annual cycle. . . (what's new about this?) and The annual cycle can also be affected.. ( has to be more specific; what difference in primary emissions?. . .) L 31: Re-write the last sentences**

The conclusion has been partially rewritten and completed.

60. **Fig 2: make the circles visible**

Done.

61. **Figs 4-5: difficult to see the differences. Maybe to plot the difference instead?**

Both figures are important for the analysis and the colortable has been chosen to highlight the differences when they exist. There are already a lot of figures and tables. Adding the difference would not add information and would increase the size of the manuscript.

62. **Fig 7: correct caption: sea salt**

Corrected.

63. **Table 7 caption: comparison MOCAGE simulations with EMEP observations**

Corrected.

---

## Author Response (AR2)

We would like to thank the Editor for his careful reading of this manuscript, and the associated remarks. The Editor's remarks are in bold and the answer are written after each point.

**Answers to Referee 1:**

**9. The bias in AOD is much smaller compared to bias for the surface stations. Can the authors discuss the reason for this?**
**Please discuss this point in the paper, and try to give a more specific response than that given to referee one.**

As already said, the AOD and the surface stations do not represent the same quantities. The AOD is a measure of the integrated column of aerosol quantities, while surface stations evaluate the aerosol concentrations at the surface only. Moreover, MODIS data cover a large area, while surface stations are limited to certain locations. When comparing to AERONET stations, the AOD is also representative of a limited amount of locations, but that are very different from the surface stations ones.
Moreover, there can be a good agreement with AODs while not with surface stations if the vertical distribution is not fully right. Also, the size distribution can affect the AOD computation, which is sensitive to the aerosol size, while the PM10 indicator is less sensitive to this aspect as long as the aerosols are smaller than 10 microns.
The manuscript has been updated in section 3.3 to add this aspect.

**Answers to Referee 2:**
**1. How the results of the present study compare with these earlier studies?**

**P10,l32: Please name the differences in sulfate you make reference to.**

The manuscript has been changed to:
Wet deposition is the main sink in Aan de Brugh et al. (2011) for sulphate, but sedimentation and dry deposition are the main sinks in this study. This can be explained by the difference of the simulated domains in both studies. The domain in our study is more southern and has high sulphate aerosol concentrations in the eastern part of the basin associated with less precipitation.

**2. Also in all tables and throughout the manuscript where reference is made to budget, it has to be clarified they concerne PM10 aerosol (or is it bulk aerosol in the model?)**

**Could authors please add this information (on use of bulk aerosol as a target) also in Figure 7 and Tables 12 and 13.**

The information has been added in figures 7 and 8 and tables 10 to 17.

**I would have prepared having budget calculations for PM10, which is more easy to reference. Could authors please discuss qualitatively, by looking at size distributions for PM species, how results would be different for PM10.**

Indeed, it would be interesting to have the same budget for PM10 if considering the air quality aspect. Nevertheless, the total mass aerosol budget is also interesting for domain such as visibility, or nutrient deposition. The simulation was set to compute the chemical production and destruction, sedimentation and dry deposition, wet deposition and burden of the total mass of aerosols. It would require to rerun the simulation by changing the outputs in order to get the budget over the PM10. Moreover, it would be very hazardous to extrapolate results to PM10.

**4. Section 2.1 MOCAGE model Information is missing on how the aerosol size is parametrized in the model. Since later in the discussion the authors refer to the aerosol sizes are taken into account in the model and how they change during atmospheric aging.**

**Authors say, that there is no effect of ageing on aerosol size in the model. Certainly , there are many processes in the model which act on aerosol size over time, such as condensation or sedimentation. May be authors want to say that effects of these processes on size distribution are small, but even this would be surprising.**

There has been a misunderstanding from our side. Indeed, by aging we were thinking of the processes that occur to a single aerosol particle, such as coagulation (this process is not explicitly represented), reaction with gas, etc. But, indeed deposition and chemical production (for secondary aerosols), and emission (for primary aerosols) processes affect the aerosol size distributions. The manuscript has been changed to be more precise on this point.

**21. Page 9, line 15-18: Nitrate aerosol partitioning to the aerosol phase is very sensitive to the aerosol pH (Guo et al., 2016) and NH4NO3 is semi volatile. How well is this computed in your model? Do you consider dust aerosol alkaline components in ISORROPIA calculations?**

**Please discuss this point also in the revised manuscript. How much more nitrate would you expect, when taking into account alkaline compounds from dust ?**
**For Pacific Ocean, Fairlie et al. (2010) address this problem and give many references on other relevant studies.**

Hodzic et al. (2006) showed that the inclusion of dust related alkaline components improves the behaviour of the model, over Europe, by reducing the PM10 bias of 6 to 10% when comparing results to selected EMEP stations. Moreover, Ansari and Pandis, (1999) showed the bias of the total nitrate aerosol is reduced by about 20% when including crustal species into the computation. Nevertheless, Fairlie et al., (2010) pointed out that the uptake coefficient usually used are too important. Hence the previous numbers might overestimated.
The manuscript has been modified to add this point. Moreover, the work to include the alkaline component in the model MOCAGE is ongoing.

Hodzic, A., Bessagnet, B., & Vautard, R. (2006). A model evaluation of coarse-mode nitrate heterogeneous formation on dust particles. *Atmospheric environment*, *40*(22), 4158-4171.
Asif S. Ansari & Spyros N. Pandis (1999). An Analysis of Four Models Predicting the Partitioning of Semivolatile Inorganic Aerosol Components, Aerosol Science and Technology, 31:2-3, 129-153, DOI: 10.1080/027868299304200

**I am quite convinced that ChArMEx provides data on alkaline compounds, or likewise data on coarse mode nitrate which are probably bound to such ions.**

On the Charmex website, the few listed speciation data such as:

http://mistrals.sedoo.fr/?editDatsId=1324&datsId=1324&project_name=ChArMEx

or

http://mistrals.sedoo.fr/?editDatsId=2&datsId=2&project_name=ChArMEx

are either not available, or not during the Charmex period (2012-2014).

**Answers to Referee 3:**
**1. The authors should make more clear the connection to the ChArMEx and the choice of the years, since they have not at all made any use of the campaign's data.**

**Even if 2012 and 2013 PM composition data from Cape Corsica are indeed not in the SEDOO**

**data base, I think they would be available from the PI (Jean Sciare), if needed. PM1 composition data from SOP2 (July, August 2013) are available and published, for example in Michoud et al., 2017. I recommend that authors use at least these SOP2 data for model evaluation.**

I send at least 5 emails to Jean Sciare to get the data, but unfortunately I never got any answer.

---

## Author Response (AR3)

Dear Editor,

Thank ou for you work on our paper. We are, at least, as disapointed as you for the fact that we didn't use ChArMEx data, but at some point we didn't want to quit the work done here. We added some text in the section 4.2.1. We also added a paragraph in the conclusion to place emphasis on the lack of crustal species interactions. Also, the budget can not be reduces to a PM2.5 budget as the sea salt interactions are included in the model. We added a sentence to be more precise about this in the section 2.1.

Best,
Jonathan Guth